# New perspectives on 'Breathomics': metabolomic profiling of non-volatile organic compounds in exhaled breath using DI-FT-ICR-MS
Madiha Malik [1,4] ✉, Tobias Demetrowitsch [2,3,4], Karin Schwarz[2,3] & Thomas Kunze [1] ✉

Breath analysis offers tremendous potential for diagnostic approaches, since it allows for easy and non-invasive sample collection. "Breathomics" as one major research field comprehensively analyses the metabolomic profile of exhaled breath providing insights into various (patho)physiological processes. Recent research, however, primarily focuses on volatile compounds. This is the first study that evaluates the non-volatile organic compounds (nVOCs) in breath following an untargeted metabolomic approach. Herein, we developed an innovative method utilizing a filter-based device for metabolite extraction. Breath samples of 101 healthy volunteers (female $n = 50$) were analysed using DI-FT-ICR-MS and biostatistically evaluated. The characterisation of the non-volatile core breathome identified more than 1100 metabolites including various amino acids, organic and fatty acids and conjugates thereof, carbohydrates as well as diverse hydrophilic and lipophilic nVOCs. The data shows gender-specific differences in metabolic patterns with 570 significant metabolites. Male and female metabolomic profiles of breath were distinguished by a random forest approach with an out-of-bag error of 0.0099. Additionally, the study examines how oral contraceptives and various lifestyle factors, like alcohol consumption, affect the non-volatile breathome. In conclusion, the successful application of a filter-based device combined with metabolomics-analyses delineate a non-volatile breathprint laying the foundation for discovering clinical biomarkers in exhaled breath.

Breath analysis has been associated with tremendous potential over the last few decades. Breath contains trace amounts of a wide range of several organic compounds[1,2]. These may serve as biomarkers indicating the presence, severity, or progression of a particular condition or disease[3]. To date, breath analysis has been studied for diagnosing and monitoring various conditions. Investigations of asthma[4,5], chronic obstructive pulmonary disease (COPD)[4], lung cancer[4], gastric and colorectal diseases[6], diabetes[7] and certain infectious diseases[8] such as COVID-19[9,10] already demonstrate the promising potential of breath analysis. Noteworthy, mentioned studies in this research field predominantly focus on volatile organic compounds (VOCs), whereas analysis of non-volatile organic compounds (nVOCs) is greatly neglected.

To obtain nVOCs, breath can be collected either via condensation methods[11] or by utilising filter-based-devices[9,12,13]. Several studies are focusing on analysing the exhaled breath condensate[10,14,15], which contains all the non-volatile substances including small molecules, ions, proteins and nucleic acids. However, the devices are complex and require a cooling tool that enables condensation of breath[11]. Moreover, ready-to-use devices utilise a polymeric, electrostatic filter to collect respiratory droplets containing nVOCs. Therefore, this approach provides a simple yet highly efficient way to acquire nVOCs, as it does not necessitate additional equipment like cooling systems, and sample collection is easily applicable. However, while these devices have been tested for specific biomarkers[9,13], their ability and capacity to simultaneously collect a wide range of nVOCs remain uncertain.

[1]Department of Clinical Pharmacy, Institute of Pharmacy, Kiel University, Kiel, Germany. [2]Institute of Human Nutrition and Food Science, Food Technology, Kiel University, Kiel, Germany. [3]Kiel Network of Analytical Spectroscopy and Mass Spectrometry, Kiel University, Kiel, Germany. [4]These authors contributed equally: Madiha Malik, Tobias Demetrowitsch. ✉e-mail: mmalik@pharmazie.uni-kiel.de; tkunze@pharmazie.uni-kiel.de

Hence, comparable to VOCs[16] analytical approaches are needed to validate the comprehensive identification of a fingerprint of nVOCs in 'healthy' breath, which could serve as a reference for investigating certain conditions or diseases. Metabolomics studies of exhaled breath (EB) have been recently described as "Breathomics"[2]. According to its definition, it involves solely the identification and measurement of volatile organic compounds (VOCs) in EB[17–21]. However, since nVOCs are undoubtedly part of the natural composition of exhaled breath[17], it might not capture the full picture of breath analysis. Hence, utilising a filter-based device, we aim to (1) characterise the 'healthy' core metabolome of exhaled breath including non-volatile organic compounds, (2) analyse gender-specific differences in metabolic pattern and (3) evaluate lifestyle effects on breath pattern such as consumption of alcohol. To our knowledge, this is the first metabolomics study evaluating the non-volatile metabolites in breath extensively.

## Results

Exhaled breath samples were collected from 50 female and 51 male volunteers between the ages of 20 and 40 with a normal BMI (19.0–25.0 kg m$^{-2}$). The metabolomics approach was conducted with ultra-high-resolution mass spectrometry. All positive controls exhibited increased intensities for the spiked metabolites, respectively. Additionally, the analysis of negative controls as well as blanks revealed only minimal traces of metabolites in the solvent or on the filter close that were close to the detection limit. All negative controls as well as blanks showed a comparable measurement profile over the entire measurement period, revealing no significant metabolites.

The obtained datasets were used to define non-volatile core metabolome and subsequently underwent evaluation for specific metabolites differentiating for gender, oral contraceptives intake, allergies and intolerances as well as life-style factors such as dietary habits (vegan, vegetarian and flexitarian), consumption of coffee or other sources of caffeine, intake of dietary supplements, consumption of alcohol, smoking and physical activity level (Table 1).

In total, the pre-processed data included 2645 chemical formulas and their putative metabolites that were present in at least 5% of all samples across the aforementioned phenotypic subgroups (Supplementary Data). To ensure the identification of the most stable metabolites specific to each subgroup and to minimise the potential interference due to individual variations, further processing focused exclusively on metabolites present in at least 15% of all samples (second filter step). After applying this filter, 1138 metabolites were selected for the statistical data analysis.

### Core metabolome

The complete set of annotated metabolites was used for characterising the non-volatile core metabolome of breath. This was achieved by using a Human Metabolome Database (HMDB)[22] 2023 annotation list that included corresponding HMDB IDs allowing the assignment of chemical classes in MetaboAnalyst 5.0[23].

The percentage coverage was calculated by dividing the number of annotated chemical formulas or metabolites by the number of the maximum potential hits. Only classes that accounted for at least 2% of the total hits were included ensuring that the represented classes had a substantial presence in the dataset. To characterise the core metabolome, the analysis focused on the main classes at the chemical level. This approach provides a comprehensive overview of the identified metabolites. Accordingly, various metabolites originating from diverse metabolic pathways present in breath were found. Various chemical classes such as amino acids and peptides as well as monosaccharides and fatty acids and conjugates, fatty acid esters and amides were detected in breath samples. Figure 1 depicts the core metabolome obtained with our analytical approach.

### Gender-specific distribution

In addition to the Human Metabolome Database (HMDB)[22], we used the Human Breathomics Database (HBDB)[2] as sources to create our own annotation lists, enabling the putative identification of additional

**Table 1 | Characteristics of the participants**

| Characteristic | Overall $n$ = 101 |
|---|---|
| Age (years) | 20-40 |
| Female | 50 |
| Intake of hormonal contraceptives | 16 |
| Male | 51 |
| Allergies and intolerances | 15 |
| Dietary habits | |
| Meat-eater/Flexitarian | 68 |
| Vegetarian | 28 |
| Vegan | 5 |
| Intake of dietary supplements | 45 |
| Physical activity level | |
| Regularly[a] | 56 |
| Occasionally[b] | 34 |
| No sports[c] | 11 |
| Stimulants | |
| Coffee | 92 |
| Smoking | 10 |
| Alcohol | |
| Regularly[a] | 53 |
| Occasionally[b] | 28 |
| No alcohol[c] | 20 |

[a]>once a week.
[b]>once a month.
[c]No event during the last 6 months.

metabolites that were previously unknown and exhibited significant gender-specific differences.

To assess and explore the sex-specific differences in metabolite profiles, the subjects were first classified according to their biological gender. These groups were then subjected to biostatistical evaluation using univariate analysis (volcano analysis), multivariate analysis (PLS-DA) and classification analysis (random forest). Using these three methodologies enables a comprehensive and robust insight regarding the interconnections and distinctions present within the samples. Moreover, the convergence of these diverse findings facilitates the process of cross-validation, particularly when examining metabolites identified as significant across two or three of the aforementioned analyses.

For the random forest analyses, the MetaboAnalyst algorithm, firstly, generated a training-dataset. In a second step, another test-dataset from the uploaded data was used to elaborate whether gender classification is successfully performed only based on metabolites. Moreover, the seeds were randomly selected. By applying the random forest approach, the algorithm was able to classify the dataset with remarkable success (Fig. 2a). All male subjects and almost all female subjects were correctly assigned to their respective group, with the exception of one woman erroneously classified as man.

Consequently, the measured out-of-bag error rate is <1%, highlighting the high accuracy of the classification. Figure 2b exhibits a scores plot from the PLS-DA analysis, demonstrating a distinct cluster separation between the two genders. The Volcano analysis of the data revealed significant gender-specific results as well (Fig. 2c). The analysis incorporates a $p$-value of <0.1 (FDR corrected) and a fold-change of >1, leading to the identification of 570 significant hits (Fig. 2c). Still, 210 significant metabolites can be found when using a $p$-value of 0.001 and a fold-change of >1.

Finally, the consistency of all the three methods was tested using a Venn diagram examining overlaps and distinctions (Fig. 2d). This approach revealed 99 metabolites as significant across all three calculations. The ten

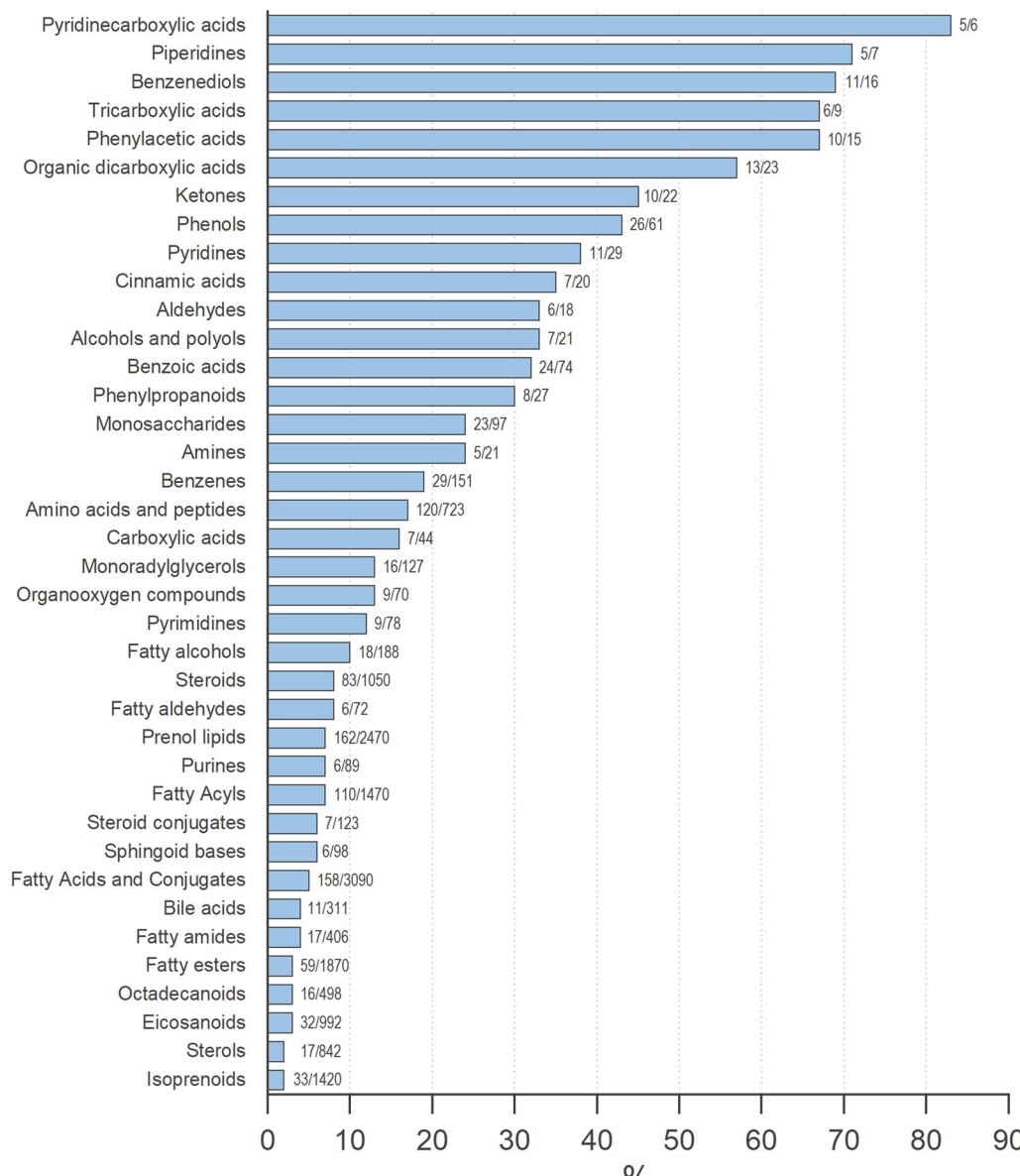

**Fig. 1 | The non-volatile core metabolome of breath classified by chemical classes.** The values displayed at the bars represent the respective number of actually detected metabolites in exhaled breath (left) versus the maximum number of potential metabolites registered in the HMDB[22] (right). The x-axis summarises the resulting fraction of the respective chemical class found in exhaled breath in percent [%].

most significant metabolites found across all three statistical tests are presented in the Supplementary Table 1. Additionally, there were no metabolites exclusively present in the VIP metabolites of the PLS-DA, while 384 and 55 metabolites were unique to the volcano analysis and random forest, respectively. The most substantial overlap, however, was observed between the random forest and volcano analysis, with 183 shared metabolites. 86 metabolites overlapped between the PLS-DA and volcano analysis. However, no metabolites were found to overlap exclusively between the random forest and PLS-DA.

Besides the classification based on the whole HMDB[22] and HBDB[2], we utilised specific gender markers from the HMDB[22] as a validation cohort. 515 metabolites were used for the re-annotation with the same MetaboScape parameters as described above. Following $t$-test calculations, 19 metabolites exhibited significant hits with a $p$-value < 0.1 (FDR adjusted) and a fold-change (FC) > 1 (Table 2).

In total, 8 metabolites exhibited the same sex-specific distribution in exhaled breath as previously published for other bio-fluids. Among those, six metabolites, including proline, tyrosine and malate, showed higher intensities in males compared to females. In contrast, two metabolites, serine and hypoxanthine, displayed higher intensities in females compared to males. However, for the remaining 11 metabolites we observed a conflicting gender-specific distribution when compared to the previously published data.

## Oral contraceptives

For further investigations on gender-specific metabolites, the female group was categorised into two subgroups regarding their use of hormonal contraceptives. 16 out of 50 women were taking oral contraceptives ("FOC"), whereas 34 were not ("FnOC").

Notably, using the Tukey HSD post-hoc test, 41 metabolites exhibited significant differences between the two female groups (FOC vs. FnOC). Pyridoxal, a derivative of vitamin B6, displayed the highest intensity among women who do not take oral contraceptives. Summarising the aforementioned analyses, Table 3 presents a selection of metabolites with significantly differing intensities identified in the distinct groups. The selection only included endogenous metabolites.

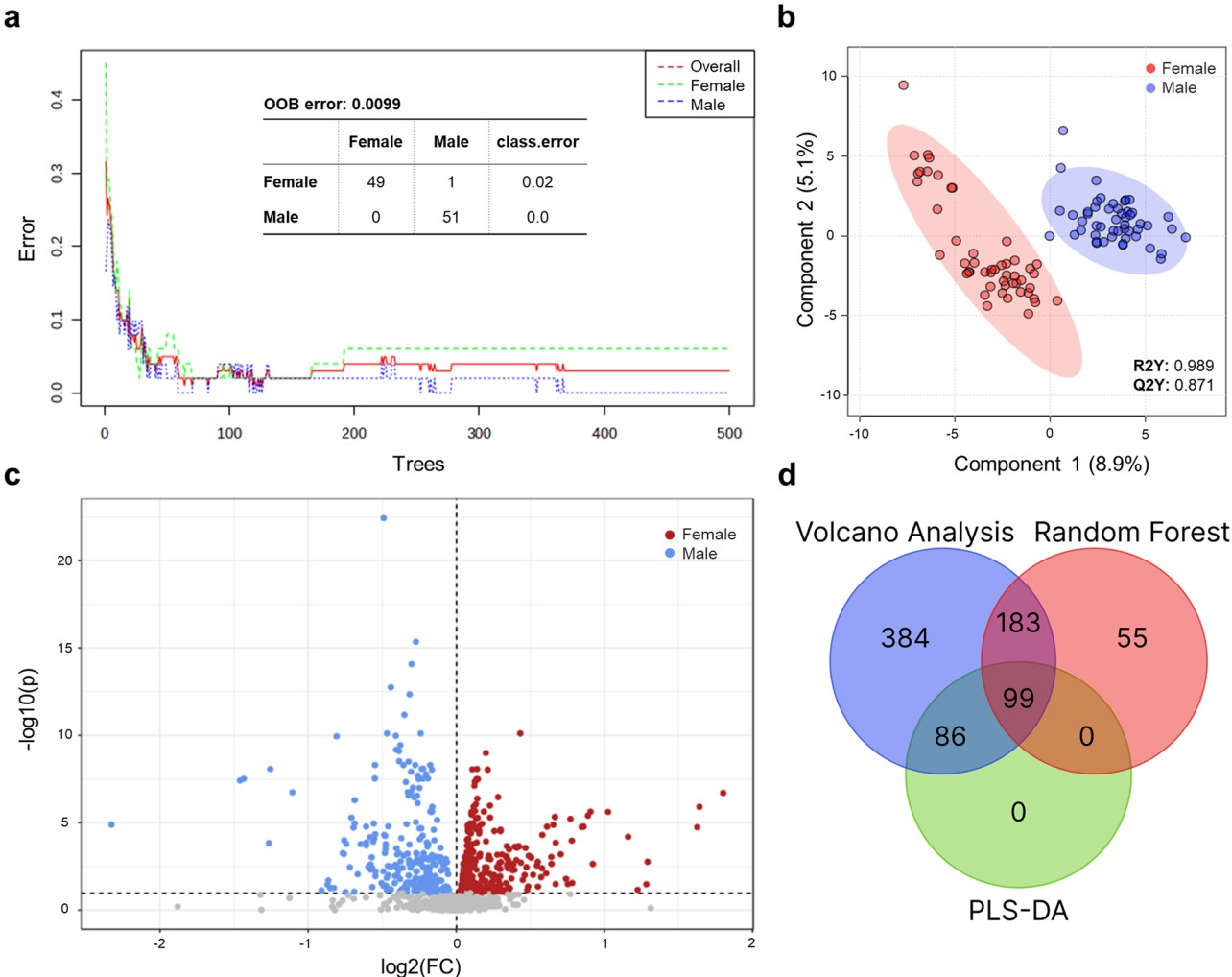

**Fig. 2 | Elaboration of gender-specific metabolic patterns using three distinct biostatistical methods. a** Random forest classification with 52 predictors and 500 trees exhibiting an out-of-bag (OOB) error of 0.0099. All males and 49 out of 50 females were accurately classified based on their metabolite profiles, respectively. **b** Scores plot of a PLS-DA analysis depicting a clear separation of metabolite profiles depending on gender with R2Y = 0.989 and Q2Y = 0.871 for five components. **c** Volcano plot depicting metabolites with a fold-change greater than 1 and a p-value lower than 0.1 (adjusted using FDR correction) with 570 metabolites significantly differing in binary gender. All calculations and analyses were carried out using MetaboAnalyst 5.0. **d** VENN diagram representing the data analysis using three different biostatistical tests. Metabolite selection was based on the specific criteria: PLS-DA VIP > 1.5, random forest Mean Dec. Acc. >0.00016, adjusted p-value of <0.1.

## Lifestyle phenotyping

In addition, we examined effects of various lifestyle factors on respective breath patterns. The subjects were instructed to avoid consuming alcohol, coffee, tobacco, and other food and beverages two hours prior to sample collection. Still, several significant metabolites were found in exhaled breath, differentiating for alcohol, coffee and tobacco consumption.

Regarding the consumption of alcohol, 20 individuals reported to abstain from alcohol, while 81 individuals indicated drinking alcoholic beverages at least occasionally. Of these 81 participants, 53 reported consuming alcohol regularly, with an average frequency of more than once a week, while the remaining 28 reported occasional alcohol consumption, occurring more than once a month (Table 1). The gender distribution among individuals practicing alcohol abstinence and those consuming alcohol were evenly balanced. Nine males and 11 females reported refraining from any alcoholic beverages, while the alcohol-consuming group counted 42 males and 39 females (Supplementary Data). When comparing alcohol-abstinent subjects to alcohol drinkers, we identified six significantly increased metabolites in the abstinent

group. Among these metabolites, four were correlated with the lipid metabolism, while the fifth metabolite, carboxymethylbutyl hydroxychroman, represented an intermediate of the vitamin E metabolism (Fig. 3).

Additionally, a metabolite with the chemical formula $C_{16}H_{22}O_3$ exhibited notably higher intensity levels in the alcohol-abstinent group. Conversely, only a single metabolite with the chemical formula $C_4H_7NO_2$ displayed elevated levels in the alcohol-consumer group when compared to the abstinent group.

Regarding coffee consumption, nine volunteers in the study abstained from consuming coffee, whereas a significantly larger group of 92 volunteers regularly consumed coffee or caffeine-containing products at least once per day. Based on the grouping and the data, only one significantly increased metabolite with the chemical formula $C_{15}H_9FO_2$ was detected.

When considering the smoking status of the individuals, we found imbalanced groups with only 10 of 101 reporting to smoke regularly. In three smokers and across all technical replicates of those individuals, a significantly increased level of the compound with the chemical formula $C_{35}H_{66}O_8$ was detected. This compound was identified as donhexocin.

**Table 2 | Gender-specific metabolites identified in exhaled breath**

| Chemical formula | Putative identity | Expression ratio (fold change) | Gender-specific distribution | |
|---|---|---|---|---|
| | | | **Herein obtained results** | **According to literature** |
| $C_{14}H_{18}N_2O_6$ | gamma-Glutamyltyrosine | 0.66 | Male > Female | Male > Female[28] |
| $C_5H_9NO_2$ | L-Proline | 0.86 | Male > Female | Male > Female[31] |
| $C_9H_{11}NO_3$ | L-Tyrosine | 0.84 | Male > Female | Male > Female[31] |
| $C_4H_6O_5$ | Malate | 0.81 | Male > Female | Male > Female[28] |
| $C_5H_{12}N_2O_2$ | L-Ornithine | 0.76 | Male > Female | Male > Female[31] |
| $C_5H_4N_4O_3$ | Uric acid | 0.66 | Male > Female | Male > Female[28] |
| $C_3H_7NO_3$ | D-Serine | 1.27 | Female > Male | Female > Male[31] |
| $C_5H_4N_4O$ | Hypoxanthine | 1.93 | Female > Male | Female > Male[28] |
| $C_6H_8O_6$ | Ascorbic acid | 0.74 | Male > Female | Female > Male[28] |
| $C_{14}H_{26}O_2$ | Myristoleic acid | 0.81 | Male > Female | Female > Male[28] |
| $C_{40}H_{80}NO_7P$ | PC(P-18:0/14:0) | 0.62 | Male > Female | Female > Male[31] |
| $C_4H_8O_5$ | Threonic acid | 0.54 | Male > Female | Female > Male[28] |
| $C_6H_{12}O_5$ | 1,5-Anhydrosorbitol | 1.19 | Female > Male | Male > Female[28] |
| $C_{17}H_{31}NO_4$ | 9-Decenoylcarnitine | 1.31 | Female > Male | Male > Female[31] |
| $C_{11}H_{12}N_2O_2$ | L-Tryptophan | 1.35 | Female > Male | Male > Female[31] |
| $C_5H_{11}NO_2S$ | L-Methionine | 2.89 | Female > Male | Male > Female[31] |
| $C_{28}H_{50}NO_7P$ | LysoPC(20:4/0:0) | 1.74 | Female > Male | Male > Female[31] |
| $C_{16}H_{31}NO_4$ | Nonanoylcarnitine | 1.23 | Female > Male | Male > Female[31] |
| $C_{10}H_{19}NO_4$ | Propionylcarnitine | 1.22 | Female > Male | Male > Female[31] |

These metabolites have been described in previous literature for other bio-fluids such as blood and urine (HMDB gender database). This table provides the gender-specific distribution of these metabolites indicating the consistency of the respective obtained distribution with the pattern observed in published literature.

**Table 3 | A selection of endogenous metabolites obtained in Tukey HSD post-hoc test for females not taking oral contraceptives (FnOC, $n = 34$) and females taking oral contraceptives (FOC, $n = 16$)**

| Chemical formula | Putative identity | Expression ratio (fold change) | Intensities |
|---|---|---|---|
| $C_8H_9NO_3$ | Pyridoxal | 1.14 | FnOC > FOC |
| $C_{11}H_{15}NO_2$ | N-Methylsalsolinol | 1.18 | FnOC > FOC |
| $C_9H_{10}O_4$ | Homovanillic acid | 1.47 | FnOC > FOC |
| $C_7H_{10}O_6$ | 3-Dehydroquinic acid | 1.24 | FnOC > FOC |
| $C_3H_7NO_4S$ | Cysteinesulfinic acid | 0.82 | FOC > FnOC |
| $C_{18}H_{32}O_2$ | (alpha-gamma) Linoleic acid | 1.37 | FnOC > FOC |
| $C_9H_{17}NO_4$ | Acetylcarnitine | 1.17 | FnOC > FOC |
| $C_{19}H_{31}NO_4$ | Dodeca-3,6,9-trienoylcarnitine | 1.52 | FnOC > FOC |
| $C_{18}H_{26}O_3$ | 4-Oxo-9,11,13,15-octadeca-tetraenoic acid | 1.88 | FnOC > FOC |
| $C_{19}H_{36}O_5$ | Diacylglycerols | 0.90 | FOC > FnOC |
| $C_{21}H_{40}O_5$ | | 0.92 | |

These metabolites were also identified in a fold change analysis. Intensities of metabolites were found significantly higher in the firstly mentioned group. Expression ratios were obtained from the fold change analysis.

However, due to the low numbers of smokers compared to non-smokers, differences were not significant.

Finally, the other univariate tests (t-test and/or ANOVA) analysis did not yield any significant results for the remaining phenotypic groups, such as dietary habits, allergies and intolerances, physical activity level, and intake of dietary supplements.

## Discussion

"Breathomics" studies refer to metabolic investigations focused on volatile organic compounds (VOCs) in exhaled breath[17–21]. However, this definition of "Breathomics" is imprecise as it disregards nVOCs, which might encompass valuable information about the breathome. Various studies have already described a diverse picture of VOCs in breath of healthy[24] as well as diseased[3–8] individuals, while missing out possibly significant pieces of the breath puzzle. To our knowledge, there is no published data evaluating the core metabolome in breath of healthy or diseased subjects, especially focusing on the presence of non-volatile components. Thus, to complete this jigsaw we present the first metabolomic profiling of non-volatile compounds in human breath. The results of this study strongly emphasise that the term "Breathomics" has to necessarily include non-volatile organic compounds. In order to get an extensive understanding of the human breath, VOCs and nVOCs need to be examined equally.

To date, the Human Breathomics Database (HBDB) has primarily presented VOCs comprising more than 900 metabolites. However, using the HBDB as a reference, we identified an overlap of just 147 metabolites, which accounts for a mere 5.5% of the total number of nVOCs identified in our dataset. Remarkably, this study unveiled additional 2509 nVOCs that had not been previously reported for human breath.

The biggest advantage of exhaled breath is its non-invasive accessibility, allowing for easy and repeatable sampling and measurements[21]. Compared to exhaled breath condensate, the use of a filter-based device[9,12,13,25] for breath collection seems to be more practicable as sampling does not require a cooling system or a condenser. Therefore, we utilised a patient-friendly filter-based device[9,12,25] (Supplementary Fig. 1) suitable for routine analytical diagnostics.

This device is meticulously designed to collect micro-particles originating from the airway lining fluid. Due to the nature of the collection process, saliva droplets might possibly contaminate breath samples. To mitigate oral fluid contamination, the mouthpiece is designed to separate saliva and larger particles from breath, allowing only micro-particles to pass

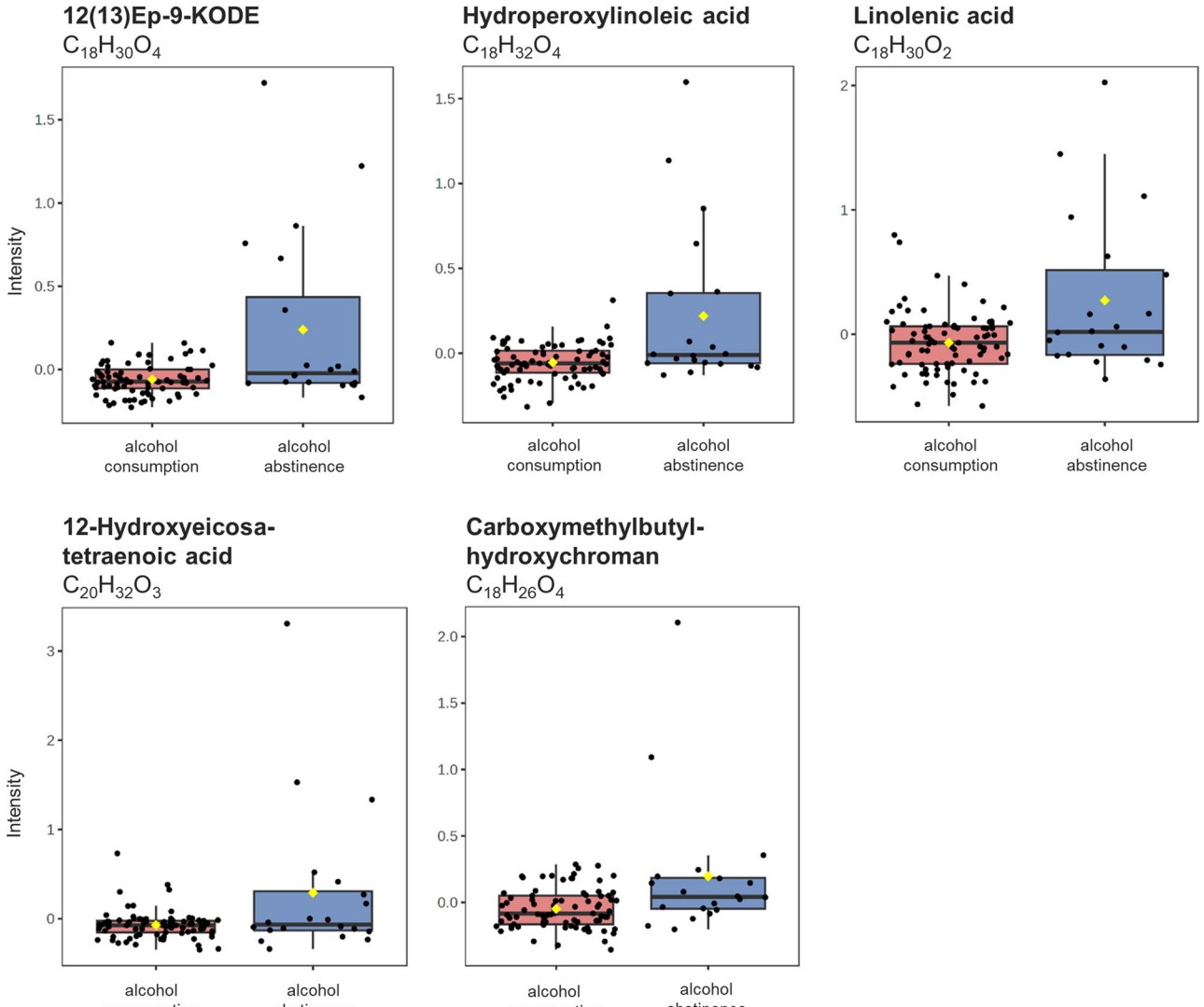

**Fig. 3 | Metabolic effect of alcohol consumption on breathome.** Box plots of five significant metabolites found in the two groups of (1) alcohol consumption (red): $n = 81$ biological independent samples compared to (2) alcohol abstinence (blue): $n = 20$ biological independent samples. Black dots represent the values from all samples. The box and whiskers summarise the normalised values with the centre line presenting the median. The mean value is indicated as a yellow diamond.

through and to be collected on the filter inside of the device[25,26]. However, it is important to note that despite these measures there may still be minimal contaminations from saliva droplets contributing to the detected metabolome.

Furthermore, the filter extraction inevitably causes loss of substances. However, it has been reported that the collection efficiency of this system used here can be assumed between 90% and 99%[9,12,25].

Accordingly, we developed an innovative analytical approach extracting metabolites from the filter-based device in multiple steps and subsequently measuring samples using FT-ICR-MS to obtain comprehensive metabolite patterns in breath. Our non-targeted metabolomics approach[27] benefits greatly from the ultra-high-resolution mass spectrometry, enabling diverse detection with an ultra-wide measurement range (65–1500 m/z), high dynamic measurement capabilities and an extremely high sensitivity distinguishing trace within femtomole range. However, the biggest challenge of this approach remains assigning identified masses to molecular structures[27]. Noteworthy, the identities presented here are putative, but the indicated chemical formulas are highly likely to be accurate.

In this study, we identified 2645 putative metabolites based on the chemical formulas, delineating the non-volatile core metabolome of breath. To ensure stability and broad hit validity, further processing

recognised metabolites present in at least 15% of all samples, yielding 1138 metabolites selected for statistical analysis. Importantly, the results are restricted to the metabolites listed in HMDB[22] 2023 and HBDB[2], i.e. other potential metabolites present in exhaled breath were therefore excluded.

According to the chemical taxonomy, the metabolites were categorised in main classes (Fig. 1) providing evidence that breath analysis can identify biochemically versatile molecules fulfilling various functions in the human body. To define the core metabolome, we used the HMDB[22] 2023 annotation list with the respective HMDB ID of the metabolites. Since not all of those IDs are deposited in MetaboAnalyst 5.0, the actual number of detected metabolites divided into the chemical classes, respectively, is most likely greater than shown here. However, more than half of the detected metabolites were used to determine the core metabolome. Based on the characteristic of exhaled breath one would assume that droplets predominantly contain hydrophilic metabolites. Surprisingly, the compounds found are greatly heterogeneous. Both hydrophilic (e.g. alcohols, amino acids or carbohydrates) and lipophilic substances (e.g. sterols, fatty acid and esters or conjugates thereof) were detected equally. The variety of chemical backbones as well as biochemical functions found among the substance classes is noteworthy.

Moreover, some of the detected lipophilic compounds serve as precursors for signal molecules, e.g. eicosanoids in prostaglandin metabolism and leukotrienes in the immune and the inflammatory response, respectively. Therefore, exhaled breath analysis might be used not only in investigations of inflammatory diseases but also for evaluation of signal transduction pathways and cellular signalling. Comparable to blood, breath demonstrably contains compounds that may be distributed in various tissues, thus, not limited to the pulmonary system.

Furthermore, this study also investigated gender-specific variations in breath composition. Considering gender-specific effects in trials consistently gain importance in biomedical research[28–30]. Accordingly, sex patterns in human metabolomics have been defined in various bio-fluids[28,31–34]. However, to the best of our knowledge, a gender-specific profile of exhaled breath has still been missing. This study aims to fill this gap. For profiling the specific gender breathome, volunteers were divided according to their self-defined binary gender. The biostatistical evaluation using univariate (volcano analysis), multivariate (PLS-DA) as well as classification (random forest) analyses unravelled interesting findings.

As shown in Fig. 2, random forest classification analysis of the measured data was proven very efficacious as all 51 male subjects and 49 female subjects were correctly assigned to their respective gender group (out-of-bag-error <1%). Only one woman was erroneously interpreted as a man. Moreover, the score plot of the PLS-DA depicted a clear cluster separation of both male and female breath samples (Fig. 2b) while the volcano plot of the dataset ascertained more than 500 significant metabolites in both groups (Fig. 2c). Analysing overlaps and differences between all three statistical methods with a Venn diagram (Fig. 2d) served as a useful tool, firstly, to evaluate the adequacy and applicability of all tests for breath analysis and, secondly, to statistically determine the most robust metabolites. Overall, 99 metabolites were identified as relevant throughout all three methods, while the volcano plot was the most sensitive analysis, identifying 384 exclusive hits. Although the distinct statistical analyses yield different results, the numerous overlapping hits still confirm their applicability. Notably, an adequate interpretation of the (most) statistically relevant metabolites generally requires a combination of univariate, multivariate and classification analyses. Nonetheless, breath samples of men and women can be distinctively differentiated proving that breath metabolites show specific gender patterns regardless of the applied biostatistical test.

For a meticulous evaluation of the specific gender pattern, the identified metabolites were re-annotated using the HMDB gender database containing 515 metabolites. According to this database, 19 gender-specific metabolites (adjusted $p$-value < 0.1, FC > 1) were obtained (Table 2). Intriguingly, comparing our results with the gender-specific pattern of exactly those metabolites described in the literature for blood or urine samples, we only found eight matches confirming the published gender-distribution. Six of those were found to be significantly higher in males and two metabolites exhibited higher intensities in females.

For instance, the results exhibit a higher intensity of uric acid in exhaled breath of men compared to women, which aligns with the fact that serum uric acid levels were reported to be higher in male[35–39]. The same pattern was reported for ornithine[31,40]. Additionally, men exhibit higher concentrations of certain amino acids in blood samples[31,32,40–43]. Our results support these findings by observing significantly higher levels of proline and tyrosine in breath samples of male subjects. However, it is also reported that women have higher concentrations of serine[31], which correlates with our result as well. Another metabolite showing higher intensities in females was hypoxanthine. At first, it seems contradictory that females exhibited lower intensities in uric acid but higher levels of hypoxanthine. However, both outcomes have been discussed in the literature and were linked to potentially elevated xanthine oxidase activities in females[44,45].

Interestingly, the eleven remaining significant metabolites exhibited an inverse intensity pattern in our data compared to their distribution described in the HMDB gender database. For example, vitamin C and its metabolites threonic acid and myristoleic acid were found to be higher in female volunteers. However, contrary to the HMDB[22], Pearson et al. support

these results implying that fasting plasma levels of vitamin C are often lower in untreated men than in women[46]. Another study exhibits the same results[47], however, making a general statement remains challenging, particularly due to the direct influence of the individual diet on the concentration of vitamin C and its metabolites.

Moreover, we were able to observe higher intensities of phosphatidylcholines and various carnitines in female subjects contradicting the HMDB assignment and other reports[41,48]. In conclusion, our findings approve the hypothesis that exhaled breath shows differing sex-specific pattern compared to other bio-fluids. Therefore, exploring breath seems promising, especially in biomarker research, as it is to some extent congruent with other bio-fluids but may also add valuable information. The finding of an indeterminate yet partially consistent sex-specific pattern in exhaled breath is likely attributable to the fact that the cells in both bio-fluids, blood and breath, originate from different barriers within the human body. Therefore, both bio-fluids are in direct contact with different types of cells. Squamous endothelial cells line the interior surface of blood vessels[49,50] while exhaled breath passes only the simple epithelial cell layer[51,52]. Thus, endothelial as well as epithelial barriers contain relatively distinctive cell types with unique functions[49,50,52]. This might cause a slightly different metabolite pattern.

A major bias in observed differences between men and women might be due to the smaller lung volume in women[53], potentially resulting in the collection of significantly varying amounts of biological material over 30 exhalations. However, breath sampling was standardised by following the manufacturer's instructions while using the device. The standardised sampling process utilised a plastic bag, which, upon filling, indicates that a sufficient breath volume has passed through the collection filter[28]. This implies that the potential effect of a smaller lung volume in women may not significantly affect the results.

Noteworthy, other covariates' effects (dietary habits, alcohol consumption, smoking, physical activity, etc.) on the gender-specific metabolic pattern appear to be negligible as the group sizes corresponding to each covariate were evenly distributed between men and women (Supplementary Data). Moreover, age or other individual morphological data such as the BMI likely play a role in sex differences and therefore may introduce a potential bias[32,34,40,54–56]. This study included volunteers with a normal BMI (19.0–25.0 kg m$^{-2}$) aged 20–40, i.e. the elderly population is not represented. Therefore, further research is necessary to explore the impact of aging and other morphological characteristics.

Besides sex differences the intake of oral contraceptives influences the metabolome. In our study, we included 16 of 50 women taking oral contraceptives daily. A fold change analysis, as well as Tukey HSD post-hoc test, revealed interesting metabolites significantly differing between women taking oral contraceptives and those not taking them. For example, the vitamin B6 derivative pyridoxal showed higher intensities in women without contraceptives. Lumeng et al. have reported the pyridoxal deficiency in women using oral contraceptives. They found plasma levels of pyridoxal phosphate with the same distribution of pyridoxal between the groups[57,58] as presented here.

Interestingly, we found metabolites of L-dopa such as homovannilic acid to be at highest level in women refusing oral contraceptives. It seems consistent that hormonal agents influence the biosynthesis as well as the metabolism of certain amino acids and neurotransmitters[58–60]. Further analysis revealed the endogenous neurotoxin methylsalsolinol and diverse lipophilic metabolites like linoleic acid to be higher abundant, while diacylglycerols showed decreased levels in females waiving hormonal agents (Table 3).

Overall, our findings unequivocally show that oral contraceptives significantly affect metabolomic patterns. However, it should be considered that the results base on a relatively small population. This influence might diminish with age or in postmenopausal women[34,40,55,56]. In accordance to Ruoppollo et al.[41] and Rauschert et al.[48], we conclude that future studies generally need to account for effects related to the intake of oral contraceptives in metabolomic approaches.

However, assessing the effects of oral contraceptives without accounting for the menstrual cycle period introduces a considerable bias into our analysis. We primarily focused on examining general effects of oral contraceptives. In order to analyse conclusive effects, future investigations must consider the exact menstrual cycle period, necessitating larger group sizes and a time-dependent analysis, given the considerable variations between menstrual cycles.

In addition to the gender-specific pattern and the intake of oral contraceptives, effects of various lifestyle factors were tested as well. Intriguingly, significant metabolites confirmed the impact of alcohol intake on the metabolomic pattern in human breath. Six significantly increased metabolites were determined in non-drinkers compared to the regular- and modest drinkers, with most of these hits being related to the lipid metabolism. One hit with the chemical formula $C_{16}H_{22}O_3$ was found significantly higher in the alcohol-abstinent group, identified as 15-oxo-5,9,11,13-pentadecatetraenoic acid methyl ester in a prior NMR study[61].

Additionally, linolenic acid, as a precursor of arachidonic acid, plays a major role for generating various relevant metabolites, including 12-hydroxyeicosatetraenoic acid (12-HETE)[62], 5,6-epoxy-8,11,14-eicosatrienoic acid (EET)[62], EP-9 KODE (an octadecanoid oxylipin) and hydroperoxylinoleic acid[63], all of which were significant hits in persons with no alcohol intake (Fig. 3). EP-9 KODE and 12-HETE are formed from linoleic acid via activation of enzymes such as lipoxygenases and cyclooxygenases[62,63] known as the first and second pathway of arachidonic acid. Furthermore, arachidonic acids can be converted to EETs via CYP epoxygenases, representing the third pathway of arachidonic acid metabolism[62]. This proves that EB analysis might be useful for monitoring changes in diverse metabolic pathways.

Research indicates that alcohol can influence fatty acid metabolism by affecting enzyme expression and activity, such as reducing 5-lipoxygenase activity[64], which is involved in HETE formation[65]. When intoxicated, the body prioritises alcohol metabolism, potentially leading to increased oxidation of alcohol and, conversely, reduced fatty acid metabolism. This might be possibly the reason for decreased availability of linolenic acid[66] and consequently lower concentrations of the associated metabolites. Another reason contributing to reduced linolenic acid synthesis might be the discussed inhibitory effect of alcohol on the delta desaturase activity[66].

Additionally, 5'-carboxy-gamma-chromanol (gamma-alpha-CHMBC), the most abundant intermediates in vitamin E metabolism[67,68], exhibited higher intensities in volunteers not consuming alcohol. Various studies have reported beneficial effects of tocotrienols, including hepatoprotective effects in patients[69] and potential reductions in hepatic triglyceride synthesis and their improved transport[70]. As alcohol consumption is known to elevate triglyceride levels[71], it may lead to decreased tocotrienol and the respective metabolite levels. Consequently, these may not effectively protect against triglyceride elevation in moderate and regular alcohol drinkers.

Moreover, comparing smokers to non-smokers there were not any significant hits due to the very small group size of smokers (10 of 101). However, we detected an increased metabolite with the formula $C_{35}H_{66}O_8$ present in three smokers, noteworthy, in all of their technical replicates. This compound's putative identity is donhexocin (acetogenin), a polyketide found in plants of Annonaceae[72]. Interestingly, it might also originate from Lasioderma serricorne, the cigarette beetle, in which polyketides play a crucial role in the synthesis of sex pheromones[73]. It is reported that acetogenins have neurotoxic potential[72,74]. This indicates that exogenous toxic substances are also detectable in EB as well.

Furthermore, we found one significantly increased metabolite with the formula $C_{15}H_9FO_2$ in those volunteers who consume coffee. It might originate from coffee itself or from the human microbiome. However, no putative metabolite has yet been described for this chemical formula, which was confirmed with an accuracy of <1 ppm and an mSigma value of 15 (a degree of isotopic fine structure). Nevertheless, it should be considered that only nine volunteers of 101 did not consume coffee or food/drinks containing caffeine. The unequal group size might have led to this statistical

result. All other lifestyle factors such as physical activity, allergies, intake of supplements and other individual diets yielded no significant results.

In general, these factors seem highly uncertain and present significant challenges in standardising the study population. To identify significant differences based on physical activity, distinguishing between not only the exercise frequencies but also the diversity of physical activities and sports could offer valuable insights. The observed limited impact of physical activity might be attributed to the small group size of individuals reporting no physical activity (11%). The same applies to allergies and intolerances, which were reported by only 15 participants. These encompassed a wide range of allergies and intolerances, including lactose, fructose, gluten, nuts, tomatoes, various fruits such as apple, pomegranate, raspberry, cherry, etc. Due to the diverse nature of these food intolerances and allergies, they may not collectively contribute to the same metabolite pattern in individuals, and the variations might not be significant enough to yield conclusive results. Moreover, the insignificant variations regarding dietary habits and dietary supplement intake could be attributed to the imposed restriction, which prohibited participants from consuming any food or beverages within two hours prior to sampling. However, to asses notable changes based on the nutrition, future studies should consider sampling immediately after food ingestion. Additionally, significant findings could be achieved by cross-referencing with diverse food and nutrient databases for metabolite annotations. Our study's findings are confined to the HMDB 2023 and HBDB lists. Therefore, metabolites not listed in both databases, which could potentially be present in exhaled breath, remained undetected.

In general, however, the study might lack statistical power to reliably assess the influence of various covariates, especially due to small group sizes. Nevertheless, these findings indicate a considerable inter- and possibly intra-individual variability concerning the aforementioned lifestyle factors. However, large-scale studies are necessary to better illustrate overarching trends.

Additionally, it should be noted that the generation of droplets could be highly variable, inter- and intra-individually, between two distinct periods of the day. However, due to the study design, breath samples were collected within a half-hour period. Consequently, the obtained results do not conclusively account droplet variability across different periods. Future studies should aim to compare sampling across multiple periods to effectively assess droplet variability, potentially revealing differing inter- and intra-individual metabolite patterns throughout the day.

Furthermore, HMDB annotation lists present a substantial weakness of this study. Utilising these may result in implausible or questionable annotated metabolites (Supplementary Data), likely due to the presence of numerous isobaric compounds that share identical chemical formulas. In these cases, MetaboAnalyst selects the first listed metabolite from the annotation list, potentially leading to a questionable assignment of certain metabolites. Therefore, although the provided chemical formulas are highly accurate, this analytical approach can only suggest putative identities. Furthermore, despite methodological variations, the greatest challenge in exhaled breath research is the standardisation of sample collection, necessitating further optimised analytical approaches.

This proof-of-concept study demonstrates a compelling metabolomic approach for exhaled breath analysis in diagnostic approaches. We depict a metabolomic profile of purportedly healthy subjects. However, patients suffering from distinctive diseases may possibly show interesting metabolite patterns, likely altering from those of healthy subjects. Therefore, future studies on exhaled breath should apply this metabolomic approach in order to examine metabolites certainly serving as potential biomarkers for various diseases, especially conditions lacking early-onset signs.

In conclusion, the findings of this study suggest expanding the definition of "Breathomics" by incorporating nVOCs. DI-FT-ICR-MS was successfully applied to delineate a non-volatile metabolome breathprint laying the foundation for the discovery of clinical biomarkers in exhaled breath. Compared to other biomaterials, exhaled breath is not only equally applicable but also easily accessible to detect a wide range of metabolites offering diverse, advantageous perspectives. Hence, metabolomic profiling

of exhaled breath could potentially disclose essential metabolites for understanding the pathogenesis of certain diseases or for establishing diagnostic breathprints. The sensible integration of shotgun DI-FT-ICR-MS into exhaled breath analysis serves as an excellent tool to achieve this goal. In general, however, evaluating sex- and age-related differences in the breathome of healthy individuals is the first step towards proposing translational applications from the bench to the bedside. This approach might therefore be applicable in personalised medicine as well.

## Methods

### Study design

We examined volunteers to develop an analytical approach for metabolomic profiling of exhaled breath (EB). The breath screening was performed utilising a filter-based device enabling the collection of EB samples. In total, 101 healthy participants (female $n = 50$, male $n = 51$) with a normal BMI (19.0–25.0 kg m$^{-2}$) aged between 20 and 40 years were recruited cross-sectional between 11 August 2022 and 19 January 2023. Each participant provided 3 replicates within one hour.

All volunteers signed an informed consent before taking part in the study. This study was conducted in accordance with the Declaration of Helsinki (as revised in 2013) and was approved (D511/22) by the local Ethics Committee of the Faculty of Medicine, Kiel University, Germany. Informed consent for research was reviewed by the data protection office of the Faculty of Medicine, Kiel University, Germany. All ethical regulations relevant to human research participants were followed.

To characterise the study population, a questionnaire was designed obtaining information about each individual's lifestyle. It included key features such as questions about nutrition and dietary habits (vegan, vegetarian, flexitarian, intolerances, allergies and intake of supplements), fitness and physical exercise (frequency), occasional or frequent intake of stimulants and medication (smoking cigarettes/nicotine, alcohol, caffeine, intoxicants and drugs).

The inclusion criteria covered specific characteristics of the individual health status. This study included only those individuals (1) not diagnosed with chronic conditions or diseases, such as diseases of the airways of the lungs, disorders of the endocrine system, metabolic disorders, as well as those (2) not suffering from an acute (contagious) disease, e.g. infection of respiratory tract like COVID-19. In addition, regular medication (except from oral contraceptives) and acute medication such as antibiotics were exclusion criteria. Moreover, patients with mental disorders and pregnant women were excluded from the study. Additionally, in order to account for potential influences of the individual nutrition on the metabolomic profile, the participants were requested to provide a detailed record of their dietary intake, listing all food and beverages consumed during the 24 h before taking part in the study.

### Exhaled Breath Specimen collection

After filling in the questionnaire, the participants performed the breath test. The EB samples were collected with a non-invasive method using a device with a polymeric electret filter (SensAbues®, Stockholm, Sweden) (Supplementary Fig. 1) as described earlier[9]. It consists of a mouthpiece, a polymeric electret filter enclosed in a plastic collection chamber, and an attached plastic bag. The mouthpiece is designed to ensure pristine breath sampling by effectively preventing any oral fluid contamination. It features spikes that act as saliva separator[25], allowing only micro-particles to pass through and to be collected on the filter inside of the device. In addition, the plastic bag inflates with a fraction of the collected air, serving as an indicator of both proper individual usage and sufficient breath volume passing through the electret filter. This ensures a consistent and standardised quantity for sampling purposes[26].

Participants were explicitly asked to avoid any intake (eating, drinking, and smoking) for two hours prior to sample collection. To obtain the EB samples, participants tidally exhaled 30 times at rest into the sampling device. One breath cycle consisted of inhaling via the nose followed by exhaling through the mouthpiece onto the filter inside of the collection device. Three samples per participant were taken each using a new device. The study staff entirely supervised the process including briefing the participants directly prior to collecting the sample. Subsequently, all samples were stored at -80 °C until extraction.

### Sample preparation

The extraction of the electret filter was performed in a similar way as described earlier[9]. The EB collection device was placed onto test tubes (Eppendorf tubes). Then, 1 mL of methanol was pipetted every ten minutes through the device in five steps to a total volume of 5 mL to extract all organic components.

All samples were dried at 45 °C in a vacuum concentrator (SpeedVac, thermo fisher scientific, Bremen, Germany). After drying, the residue was re-suspended in 350 µl methanol/H$_2$O (1:1, v/v). Subsequently, all samples were agitated for 60 minutes in an orbital shaker.

Additionally, 6 negative controls and six positive controls were prepared. The negative controls were prepared using non-sampled new collection devices. For a positive control a solution of a mixture of phenylalanine, tyrosine, lysine, glutathione, glucose, palmitic acid and dipalmitoyl phosphatidyl choline with a concentration of $2.0 \times 10^{-5}$ mol/l, respectively, was prepared in methanol/H$_2$O (1:1). 100 µl of this mixture were pipetted onto the filter in the collection device, containing 200 pmol of each substance. Subsequently, the filters were extracted in the same way as the exhaled breath samples. For analysis, each sample was diluted 1:10 with methanol/H$_2$O (1:1). The time interval between sampling and analysis was up to six months. After participants were recruited between August 2022 and January 2023, all samples were prepared in batches within four weeks. Subsequently, mass analysis was promptly conducted without any interruptions over a continuous period of 2 weeks. Prior to analysis, all sample vials were stored at −80 °C.

### DI-FT-ICR-MS measurements

The mass spectrometric analysis was performed with a FT-ICR-MS system (7 Tesla, Solari-XR-X, Bruker, Bremen, Germany) linked to an Infinity 1260 HPLC in the direct injection mode (Agilent, Waldbronn, Germany). A methanol/H$_2$O solvent (1:1, v/v) was used as eluent with a flow rate of 0.01 ml/min. 80 µl of each sample were injected without chromatographic separation. The FT-ICR-MS was equipped with a 2-omega cell, which provides higher spectra rates (halved acquisition time with similar resolution compared to a 1-omega cell) and improved precision of attained m/z signals (<1 ppm). The mass resolution was approximately 488,000 by mass 400 m/z.

An electrospray ionisation source was used for sample ionisation in both modi (positive and negative). Two different measurement methods, one for the detection of ultra-small molecules and the other for small molecules) were employed to obtain a total detection range from 65 to 1500 m/z.

The main parameters set for the mass spectrometry were: (1) nitrogen as drying gas (4 L/min at 200 °C), (2) nitrogen as nebuliser (1 bar), (3) time-of-flight section: 0.35 (for the ultra-small method with a mass range of 65-300 m/z) or 0.6 ms (for the small molecule method with a mass range of 95–1500 m/z), (4) quadrupole mass: 80 or 120 m/z (depending on method), (5) detector sweep excitation power of the ICR-cell for both methods: 18%, (6) average scans: 292 with an accumulation time of 0.5 s or 268 with an accumulation time of 0.1 s (depending on method).

Before analyses, the source area was cleaned with isopropanol/H$_2$O (1:1, v/v) and the ICR-cell was calibrated with sodium trifluoroacetate to an accuracy of <0.5 ppm.

The monitoring of measurement stability was carried out with quality control (QC) samples representing a pool of all samples. To obtain QCs, 10 µl of each sample were transferred to a collective sample vial and subsequently processed identically compared to the analytical samples. The QC sample was measured in the beginning, after 12 and 24 samples, respectively, and at the end of each method following a modified approach of Jensen-Kroll et al.[75]. In total, 34 QC samples were measured for each method.

Observing the QC samples allowed assessing the stability of the system, as well as the dilution and stability of the samples.

## Sample evaluation

Raw data was imported to MetaboScape 2021b (Bruker, Bremen, Germany) and recalibrated by means of a local calibration list, which contains >60 different matrix metabolites (e.g. amino acids, sugar, lipids) to an accuracy of <1 ppm. Each method was imported and processed separately.

For metabolite annotation, we utilised two databases: the Human Metabolome Database 5.0 (HMDB)[22] and the Human Breathomics Database (HBDB)[2]. The HMDB is an extensive open-access database that provides details about endogenous as well as exogenous small molecule metabolites present in the human body. It offers comprehensive data on metabolite structures, chemical properties, biological functions, concentrations, and their correlations with various diseases[22]. In contrast, the HBDB serves as a knowledge database, encompassing more than 900 metabolites reported in 2766 publications associated with human breathomics. It is currently the most comprehensive database predominantly focusing on volatile organic compounds found in human exhaled breath[2]. A download version of the HMDB[22] 2023 as well as the HBDB[2] were used with a mass error of <1 ppm and isotopic fine structure error of <300. Only sodium, potassium and chloride were allowed as chemical adduct because of the direct injection analyses without a chromatography separation.

The calculated bucket tables were exported to R 4.3. All data was normalised by a developed PQN and by means of the measured QC samples[76]. Additionally, different batches were corrected by applying the PQN and median normalisation. The normalised data originating from the two-method dataset as well as from both ionisation modes were merged to create a single dataset covering the complete mass range. This merging process resulted in a combined annotation table. As part of data merging process, the occurrence of a metabolite was counted in both ionisation modes across all samples, respectively. Subsequently, the ionisation mode values with the highest number of detections of a specific metabolite across all samples were chosen for further data analysis, while the alternate mode values with lower numbers were disregarded. For instance, if a metabolite was detected 91 and 81 times in the positive and negative mode, respectively, the algorithm proceeded to retain the values from the positive ionisation mode. In cases where a metabolite was detected in an equal number of samples in both modes, the total intensity served as a secondary criterion for evaluation.

To eliminate duplicate metabolites, we followed the method outlined by Brix et al[76]. with the following rule: The feature displaying the highest overall intensity across all samples was consistently retained for further analysis. Duplicates with lower scores were subsequently removed. This approach allowed eliminating duplicates for both ionisation techniques effectively, including the overlapping mass range of 95-300 Dalton.

The next step involved the data reducing of the three replicates. For this, a metabolite had to be present in at least two samples of the three replicates per volunteer. If so, the mean was calculated based on the two or three obtained intensities. If a metabolite was only present in one sample per three replicates, the metabolite was set as "not detected" for this volunteer. Finally, volunteers were grouped by applying different filters based on the phenotype data. Accordingly, we characterised volunteers based on gender assignment, smoking status, dietary habits, physical activity level, alcohol consumption, supplement intake, and oral contraceptive use.

## Statistics and reproducibility

The filtered and grouped data was transferred to MetaboAnalyst 5.0[23]. It was uploaded using the peak intensities with the samples organised in columns (unpaired). To address the missing (zero) values, a fixed value equal to 1/5 of the limit of detection ($10^6$ counts for all metabolites) was used. Moreover, metabolites that were not present in at least 15% of all samples were removed to avoid noise peaks. The filter value was adapted considering the size of the phenotype groups.

Regarding the statistical settings, we utilised the median intensity value. Following that, the data underwent median normalisation. To achieve a normal distribution, all the data were log-transformed and Pareto-scaled.

To compare two groups such as smokers and non-smokers, we performed a volcano analysis. The log fold-change was set to >1 (indicating a double effect size between the groups), and the adjusted $p$-value was set to <0.1 (false discovery rate (FDR) adjusted for multiple testing). Furthermore, to enable comparisons between multiple groups such as vegan, vegetarian or flexitarian, we applied a multiple ANOVA combined with a Tukey honestly significant difference (HSD) test for FDR correction. The significance threshold was set at a $p$-value of <0.1.

We performed a multivariate analysis using a partial least squares discriminant analysis (PLS-DA). This allowed us to generate a score-plot and determine the variable importance for projection (VIP) features, which highlight pattern differences.

For automated sample classification, a random forest algorithm, which is a software part of MetaboAnalyst 5.0, was performed. To determine the optimal number of predictors for the random forest analysis, the square root of the total number of putatively identified metabolites in the dataset was utilised (before data processing: $n = 2656$, 52 predictor variables). Additionally, a maximum of 1000 decision trees were allowed. By selecting these parameters, we were able to determine the out-of-bag (OOB) error and the classification error (class.error). Random training samples were picked automatically by MetaboAnalyst random forest algorithm.

Finally, the gender classification was used as an illustrative example to highlight the results obtained from the three distinct biostatistical methods. The significant results obtained from the volcano analysis, the VIP metabolites from PLS-DA, and the significant results of the random forest were combined in a Venn diagram. The diagram allows for the assessment of the overlap between the three methods.

## Reporting summary

Further information on research design is available in the Nature Portfolio Reporting Summary linked to this article.

## Data availability

The (pre-processed) raw dataset, population characteristics and source data behind the graphs are available in the Supplementary Data. All other data are available from the corresponding authors on reasonable request. Further inquiries can be directed to mmalik@pharmazie.uni-kiel.de or tkunze@-pharmazie.uni-kiel.de.

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

## Acknowledgements
The authors sincerely thank all volunteers for participating in this study.

## Author contributions
Madiha Malik contributed to the design and conception of the study, recruited all volunteers, collected and prepared all samples, carried out the experiments, prepared the figures, performed biostatistical analysis and wrote the manuscript. Tobias Demetrowitsch carried out the experiments, performed the biostatistical analysis, prepared the figures and wrote the manuscript. Karin Schwarz was involved in designing and conceptualising the study. Thomas Kunze conceived, designed, and supervised the study. Madiha Malik and Tobias Demetrowitsch contributed equally to this manuscript. All authors revised and approved the manuscript.

## Funding

## Competing interests
The authors declare no competing interests.
