## [Peer Review File · Communications Biology]

Reviewers' comments:

Reviewer #1 (Remarks to the Author):

The study by Madhia et al. expands the definition of breathome by monitoring the non-volatile organic compounds (nVOCs) in the exhaled breath of healthy volunteers on which limited studies have been attempted so far in the literature. Until now breathome included majorly on the volatile organic compounds (VOCs) and this study has characterized the nVOC parts of the exhaled breath. Reports demonstrated that exhaled breath VOCs show sex specific patterns. This report elegantly demonstrated that certain nVOCs show sex-specific differences in their concentration. In addition, this study also showed that female healthy volunteer's consuming oral contraceptives showed differences in the exhaled breath nVOC composition. However, lifestyle choices like alcohol and coffee consumption had limited impact on the nVOC composition. The present study shows that nVOCs identified in the exhaled breath consists of varied chemical classes which might be contributed by multiple host tissues and may not be restricted only to the pulmonary system. Therefore, exhaled breath-based metabolomics studies focusing on nVOCs could be useful to identify disease specific biosignatures for early non-invasive diagnosis of various pathophysiological conditions.

This is an important report and should be favourably considered for publication in this journal after incorporating the following comments.

Major Comments

1. Details of the participants in terms of age, body mass index, sex, smoking status, dietary habit, alcohol and caffeine/coffee consumption seems speculative. These parameters should be tabulated in the manuscript and shared with the reader.
2. The sex distribution of alcohol abstinence and alcohol consuming groups should be added. Details on the average amount of alcohol consumed by the members in the specific groups, in the manuscript, will provide critical details for future comparative analysis.
3. The details on the coffee dose consumed by the coffee consumption group could be incorporated in the manuscript which might explain the dose on which it has limited influence on the nVOC composition of the exhaled breath.
4. Possible explanations which might have contributed to the observed insignificant variations based on the dietary habits, allergies and intolerances, physical activity level, and intake of dietary supplements could be expanded in the discussion section.

Minor Comments

1. The HMDB and HBDB terms need expansion.
2. A higher resolution Figure 2 could be incorporated to make the texts more legible.
3. Ln 117–123 could be shifted after Ln 135.
4. Please explain "diet protocol" in Ln 452.

Reviewer #2 (Remarks to the Author):

The authors report the results of a study conducted on 101 healthy volunteers, from whom expired breath samples were collected using a passive collection device containing an electrostatic air filter,

used to collect submicron bioaerosol particles. The filter content was then analyzed using a direct injection FT-ICR-MS method, and the data were primarily processed using online tools (MetaboAnalyst, HMDB, HBDB) to describe the main metabolites identified in these samples and to investigate expression differences based on covariates of interest (gender, dietary habits, smoking, etc.). The authors describe sets of differentially expressed metabolites between men and women, as well as based on exposure to oral contraceptives, alcohol, or tobacco.

The manuscript is well-constructed, easy to read, but some issues need addressing before considering it for publication, particularly in clarifying how certain potential major biases were accounted for.

Major comments:

1. The device is used to sample the particulate phase of expired breath, consisting, among other things, of particles originating from the airway lining fluid. The potential contribution of saliva droplets to the detected metabolome should be discussed. The generation of droplets can be highly variable, either between individuals or, for the same individual, between two distinct periods of the day. Since three consecutive samples were taken from each volunteer, an assessment of intra-individual variability would be of great interest to evaluate the interpretation of results obtained in future studies focusing on pathological states. The numbers of variables present in 1, 2, or 3 of the 3 replicates/patients and the CVs of expression levels should be reported.

2. Gender effect: a major bias for observed differences between men and women could be related to the smaller lung volume in women, leading to the collection of significantly different amounts of biological material during the 30 expirations (LoMauro A, Aliverti A. Sex differences in respiratory function. *Breathe* (Sheff). 2018 Jun;14(2):131-140. doi: 0.1183/20734735.000318), especially in the absence of a standardized method for sample quantity. SensAbues devices usually include a plastic bag inflating with a fraction of the air collected, serving as an indicator of the sampling volume. Was this bag used during sampling? If not, what ensures the comparability of samples from different participants? Does the gender effect persist if data are corrected for participants' morphological data (height, BMI)?

3. The authors chose to present and interpret their results by providing annotations of metabolites for each detected feature, suggesting predominant chemical families for the detected metabolites. Even though they take the precaution of specifying that the annotations are only putative, as the proposed technique allows for annotation level 2 according to MSI, there are a considerable number of features in the supplementary dataset with evidently incorrect annotations, both due to the proposed identity of metabolites and the number of samples in which they are expressed. This highlights the weaknesses of the method used and raises doubts about the quality of the annotation of the entire dataset and the interpretation that can be made, especially for chemical families and metabolic pathways. As an example, here is a non-exhaustive list of some metabolites for which annotation is questionable: mustard gas, which is a chemical warfare agent; Phosphoramidate mustard, which is a metabolite of a cytotoxic anticancer drug; 4-Hydroxypropofol, which is a metabolite of an anesthetic; (R)-Amphetamine; Valproic acid glucuronide, a metabolite of an antiepileptic; Valaciclovir, an antiviral; Secobarbital, a sedative; Psilocybin, a hallucinogen; Olamofloxacin, Propofol glucuronide; Dolasetron; Desethylchloroquine; Sparfloxacin; Loperamide; Antibiotic X 14889A; Ketotifen; Tiotropium; Antibiotic X 14889D; Risperidone; Perindoprilat; Azilsartan medoxomil; Cediranib; S-6-Hydroxywarfarin; Lorazepam; Viloxazine; 4'-Azidocytidine; Ibuprofen glucuronide; Mibefradil; Clofibrate; (R)-Ruxolitinib; Terbutaline; Tiagabine; Fluticasone 17 β -Carboxylic Acid; Spironolactone; simvastatin hydroxy acid; Taribavirin; Olodaterol; Protriptyline; Bepridil; Ispinesib; Siguazodan; Almorexant; Barbituric acid; Reproterol; cycloguanil; Cetirizine; Flunarizine; Irbesartan; Daprodustat; Imatinib; Lanabecestat; Metoprolol; Chlorambucil; Tafluprost; Ulixertinib; Vardenafil...

4. Quantitative aspects: Procedures for data normalization and standardization, especially to enable inter-individual comparisons, are insufficiently detailed. The effects of normalization using the PQN method could be shown in additional data (without vs. with normalization, on the total signal).

5. Signal processing: How were data from negative controls exploited? Was a filter applied to retain metabolites for volunteers only if their expression was significantly different from that of the blanks? How many metabolites were detected in the blanks? What were the results of the analyses of positive

controls (spiked samples)? Was a correction applied for different batches? How were results from analyses in positive and negative ionization modes managed and concatenated (number of metabolites detected in each mode, degree of overlap, number of metabolites eliminated during the data concatenation step)? How were signals corresponding to isotopes treated?

6. Missing values: "To address the missing (zero) values, a fixed value equal to 1/5 of the limit of detection was used." What are the detection limits for each compound, and how were they determined?

7. Data filter: The complete dataset had 2656 chemical formulas, and a filter was applied to keep only variables present in at least 15% of all samples, resulting in 1139 metabolites. However, in the Excel spreadsheet of results, at least 1897 variables are expressed in more than 15% of volunteers.

8. Gender and other covariates' effects: Were the group sizes corresponding to each covariate (tobacco, dietary habits, etc.) evenly distributed between men and women?

9. Oral contraceptives' effect: Analyzing based on oral contraceptive use as a binary variable makes little sense without considering the menstrual cycle period, as its effects on the metabolome have been described, and without confirming contraceptive use in the 48 hours preceding the sampling.

10. A "limitations" paragraph should be added to the discussion to address these various points, as well as the likely lack of statistical power to reliably assess the contribution of different covariates.

Minor comments:

- The methods indicate that "Participating required a diet protocol." Which protocol?
- PLS-DA for gender effect: Model metrics (R2Y, Q2Y) should be added to the graph.
- During what period were the analyses conducted? What were the time intervals between sampling and analysis?
- In tables 1 and 2, expression ratios between different groups should be indicated.
- The labels of the axes in the graphs are almost illegible.
- A number of metabolites (DG(11M3/9D3/0:0)DG(9D3/1 ; N-[(4E,8Z)-1,3-dihydroxyoctadeca-4,8-dien-2-yl]hexadecanamide 1-glucoside ; 3-(4-(2-Dimethylamino-1-methylethoxy)phenyl)-1H-pyrazolo(3,4-b)pyridine-1-acetic acid; Austalide F ; Austrobailignan 7; Bakuchiol ; CE(11:1D3) ; Ecadotril ; Glutaminyityrosine; Pratosartan; Udenafil) are not present in any sample. How can this data be found in the table?
- Consider shortening the discussion section.

REVISION NOTES

We would like to express our sincere thanks to the reviewers for their effort. Your feedback has been invaluable and the comments were truly helpful to optimize the work. In the following, we like to address the concerns raised by responding point-by-point. Additionally, all changes made in the manuscript are highlighted in yellow.

Reviewer #1:

Major Comments

- 1. Details of the participants in terms of age, body mass index, sex, smoking status, dietary habit, alcohol and caffeine/coffee consumption seems speculative. These parameters should be tabulated in the manuscript and shared with the reader.**

The manuscript now incorporates a table entitled '*Table 1: Characteristics of the participants*' detailing the main characteristics of the participants. For more comprehensive information on each participant's specific characteristics, an updated Excel sheet is available in the Extended Data file named 'Revised Non-Volatile Metabolites in Exhaled Breath_full'.

Additionally, we have specified that only participants with a normal Body Mass Index ranging between 19.0 and 25.0 kg m⁻² were included in the study. The statement '*We only included participants with a normal BMI (19.0-25.0 kg m⁻²).*' was added in the Results section and within the subsection on Study Design in the Methods.

The tables previously labelled as 'Table 1' and 'Table 2' have been renamed to 'Table 2' and 'Table 3', respectively. The updated table numbers within the manuscript have been highlighted in yellow for easy identification of the changes made.

- 2. The sex distribution of alcohol abstinence and alcohol consuming groups should be added. Details on the average amount of alcohol consumed by the members in the specific groups, in the manuscript, will provide critical details for future comparative analysis.**

We added details regarding the average amount of alcohol consumed, along with a comprehensive gender distribution within both groups in Results under the subsection 'Lifestyle Phenotyping'.

"Of these 81 participants, 53 reported consuming alcohol regularly, with an average frequency of more than once a week, while the remaining 28 reported occasional alcohol consumption, occurring more than once a month (Table 1). The gender distribution among individuals practicing alcohol abstinence and those consuming alcohol were evenly balanced. Nine males and 11 females reported refraining from any alcoholic beverages, while the alcohol-consuming group counted 42 males and 39 females (Extended Data)."

3. The details on the coffee dose consumed by the coffee consumption group could be incorporated in the manuscript which might explain the dose on which it has limited influence on the nVOC composition of the exhaled breath.

An accurate measurement of coffee dosage per individual presents challenges due to insufficient specification, variations in participants' consumption of diverse beverages with differing caffeine concentrations, and the varying ingredients and diverse quality among different types of coffee. However, when analysing the general consumption pattern based on the list of food and beverages provided (see 4. in Minor Comments), participants indicated at least a regular consumption of coffee on a daily basis.

This piece of information was added in the in Results under the subsection 'Lifestyle Phenotyping':

“Regarding coffee consumption, nine volunteers in the study abstained from consuming coffee, whereas a significantly larger group of 92 volunteers regularly consumed coffee or caffeine-containing products at least once per day.”

Moreover, the limited impact observed might stem from two potential factors: the participants were not allowed to consume any drinks or food within 2 hours prior to sampling, and the group of non-coffee drinkers was notably small, consisting of only 9 individuals. These considerations could offer possible explanations for the absence of significant findings, as further discussed in 4. (see below).

4. Possible explanations which might have contributed to the observed insignificant variations based on the dietary habits, allergies and intolerances, physical activity level, and intake of dietary supplements could be expanded in the discussion section.

Thank you. Possible explanations for the absence of significant findings observed in the abovementioned subgroups were expanded in the discussion section:

“In general, these factors seem highly uncertain and present significant challenges in standardizing the study population. To identify significant differences based on physical activity, distinguishing between not only the exercise frequencies but also the diversity of physical activities and sports could offer valuable insights. The observed limited impact of physical activity might be attributed to the small group size of individuals reporting no physical activity (11%). The same applies to allergies and intolerances, which were reported by only 15 participants. These encompassed a wide range of allergies and intolerances, including lactose, fructose, gluten, nuts, tomatoes, various fruits such as apple, pomegranate, raspberry, cherry, etc. Due to the diverse nature of these food intolerances and allergies, they may not collectively contribute to the same metabolite pattern in individuals, and the variations might not be significant enough to yield conclusive results. Moreover, the insignificant variations regarding dietary habits and dietary supplement intake could be attributed to the imposed restriction, which prohibited participants from consuming any food or beverages within two hours prior to sampling. However, to assess notable changes based on the nutrition, future studies should consider sampling immediately after food ingestion. Additionally, significant findings could be achieved by cross-referencing with diverse food and nutrient databases for metabolite annotations. Our study's findings are confined to the HMDB 2023 and HBDB lists. Therefore, metabolites not listed in both databases, which could potentially be present in exhaled breath, remained undetected. Nevertheless, these findings indicate a considerable inter- and possibly intra-individual variability concerning the aforementioned lifestyle factors. However, large-scale studies are necessary to better illustrate overarching trends.”

Minor Comments

1. The HMDB and HBDB terms need expansion.

We incorporated information regarding the two databases that were used for metabolite annotation. Both terms are elucidated within the ‘Sample evaluation’ subsection under the ‘Methods’ section.

"For metabolite annotation, we utilized two databases: the Human Metabolome Database 5.0 (HMDB) and the Human Breathomics Database (HBDB). The HMDB is an extensive open-access database that provides details about endogenous as well as exogenous small molecule metabolites present in the human body. It offers comprehensive data on metabolite structures, chemical properties, biological functions, concentrations, and their correlations with various diseases. In contrast, the HBDB serves as a knowledge database, encompassing more than 900 metabolites reported in 2766 publications associated with human breathomics. It is currently the most comprehensive database predominantly focusing on volatile organic compounds found in human exhaled breath."

Additionally, we specified the abbreviations ‘HMDB’ and ‘HBDB’ where these terms were first introduced in the text. Furthermore, we removed instances of the word ‘database’ throughout the manuscript where it was redundant in phrases such as ‘HMDB database’ or ‘HBDB database’, as the term database is already included in their names.

2. A higher resolution Figure 2 could be incorporated to make the texts more legible.

Higher resolution Figure 2 has been included in the manuscript.

3. Ln 117–123 could be shifted after Ln 135.

Thank you for suggesting this. Shifting the paragraph (lines 117-123) after line 135 makes much more sense and improves the readability.

4. Please explain “diet protocol” in Ln 452.

Thank you for pointing this out. The wording ‘diet protocol’ might be misleading. The participants were asked to write down the individual intake of any food and beverages over the last 24 hours. Each participant provided a list consisting of meals and drinks they had consumed. The sentence in line 452-453 “*Additionally, participating required a diet protocol of the last 24 hours to account for potential influences of the individual nutrition on the metabolomic profile.*” was changed to:

Additionally, in order to account for potential influences of the individual nutrition on the metabolomic profile, the participants were requested to provide a detailed record of their dietary intake, listing all food and beverages consumed during the 24 hours before taking part in the study.

Reviewer #2:

Major Comments

1. **(a) The device is used to sample the particulate phase of expired breath, consisting, among other things, of particles originating from the airway lining fluid. The potential contribution of saliva droplets to the detected metabolome should be discussed.**

Thank you for pointing this out. The following paragraph was incorporated in the discussion. In addition, we described the collection device in more detail in the methods (also see 2.)

Discussion:

"This device is meticulously designed to collect micro-particles originating from the airway lining fluid. Due to the nature of the collection process, saliva droplets might possibly contaminate breath samples. To mitigate oral fluid contamination, the mouthpiece is designed to separate saliva and larger particles from breath, allowing only micro-particles to pass through and to be collected on the filter inside of the device. However, it is important to note that despite these measures there may still be minimal contaminations from saliva droplets contributing to the detected metabolome."

Methods:

"It consists of a mouthpiece, a polymeric electret filter enclosed in a plastic collection chamber, and an attached plastic bag. The mouthpiece is designed to ensure pristine breath sampling by effectively preventing any oral fluid contamination. It features spikes that act as saliva separator²⁶, allowing only micro-particles to pass through and to be collected on the filter inside of the device. In addition, the plastic bag inflates with a fraction of the collected air, serving as an indicator of both proper individual usage and sufficient breath volume passing through the electret filter. This ensures a consistent and standardized quantity for sampling purposes²⁷."

- (b) The generation of droplets can be highly variable, either between individuals or, for the same individual, between two distinct periods of the day. Since three consecutive samples were taken from each volunteer, an assessment of intra-individual variability would be of great interest to evaluate the interpretation of results obtained in future studies focusing on pathological states. The numbers of variables present in 1, 2, or 3 of the 3 replicates/patients and the CVs of expression levels should be reported.**

Thank you for addressing this question.

We fully agree with you that droplets can be highly variable both inter- and intra-individually, across two distinct periods of the day. However, owing to our study design, each subject was sampled three times within a half-hour interval to capture a comprehensive metabolome of healthy individuals.

Indeed, repeated breath sampling of volunteers did result in slightly different metabolite profiles. Therefore, we decided to combine the information obtained from each of the three biological replicates. In order to validate the replicates, we used a heuristic filter aiming to mitigate potential variations within individuals (such as prolonged exhalation, dry mouth, lung irritation due to sampling, etc.), thereby reducing the number of 'false positives'. The rule states that in a triplicate measurement, a metabolite must be detected in at least two samples for its intensity to be determined by averaging. If a metabolite appears in only one of the three samples, the metabolite is ignored for all further processing and statistical calculations. Consequently, its intensity is set to zero for all three samples (= false positive).

Since we filtered and combined the data obtained from all replicates, we did not focus on the assessment of intra-individual variations within the study population. In contrast, this data validation rather facilitated the identification of the core metabolome of healthy volunteers. However, due to the study design, we did not collect breath samples at different times of the day. Therefore, we cannot make any definitive statements regarding droplets variability at different times. Future studies should aim to compare sampling across various time periods to assess droplet variability, potentially revealing differing inter- and intra-individual metabolite patterns throughout the day. We addressed this as part of the limitations of our study (see 10.).

2. Gender effect: (a) a major bias for observed differences between men and women could be related to the smaller lung volume in women, leading to the collection of significantly different amounts of biological material during the 30 expirations (LoMauro A, Aliverti A. Sex differences in respiratory function. *Breathe* (Sheff). 2018 Jun;14(2):131-140. doi: 0.1183/20734735.000318), especially in the absence of a standardized method for sample quantity. (b) SensAbues devices usually include a plastic bag inflating with a fraction of the air collected, serving as an indicator of the sampling volume. Was this bag used during sampling? If not, what ensures the comparability of samples from different participants? (c) Does the gender effect persist if data are corrected for participants' morphological data (height, BMI)?

(a) Thank you for addressing this point. We have discussed this potential bias in the manuscript:

"A major bias in observed differences between men and women might be due to the smaller lung volume in women⁵², potentially resulting in the collection of significantly varying amounts of biological material over 30 exhalations. However, breath sampling was standardized by following the manufacturer's instructions while using the device. The standardized sampling process utilized a plastic bag, which, upon filling, indicates that a sufficient breath volume has passed through the collection filter²⁷. This implies that the potential effect of a smaller lung volume in women may not significantly affect the results."

(b) We followed the manufacturer's instructions while using the device, which included utilizing the plastic bag as specified. To provide further clarity, we included specific details about the device in a paragraph within the method section:

"It consists of a mouthpiece, a polymeric electret filter enclosed in a plastic collection chamber, and an attached plastic bag. The mouthpiece is designed to ensure pristine breath sampling by effectively preventing any oral fluid contamination. It features spikes that act as saliva separator²⁶, allowing only micro-particles to pass through and to be collected on the filter inside of the device. In addition, the plastic bag inflates with a fraction of the collected air, serving as an indicator of both proper individual usage and sufficient breath volume passing through the electret filter. This ensures a consistent and standardized quantity for sampling purposes²⁷."

(c) We only included participants with a normal BMI (19.0-25.0 kg m⁻²) between 20 and 40 years. Therefore, we did not correct the data using any morphological data, as we conducted the study with a homogenous study population. To discuss this limitation, we made an amendment to this following paragraph in the discussion with changes highlighted in yellow.

"Moreover, age or other individual morphological data such as the BMI likely play a role in sex differences and therefore may introduce a potential bias^{31,33,39,52-54}. This study included volunteers with a normal BMI (19.0-25.0 kg m⁻²) aged 20 to 40, i.e. the elderly population is not represented. Therefore, further research is necessary to explore the impact of aging and other morphological characteristics."

3. The authors chose to present and interpret their results by providing annotations of metabolites for each detected feature, suggesting predominant chemical families for the detected metabolites. Even though they take the precaution of specifying that the annotations are only putative, as the proposed technique allows for annotation level 2 according to MSI, there are a considerable number of features in the supplementary dataset with evidently incorrect annotations, both due to the proposed identity of metabolites and the number of samples in which they are expressed. This highlights the weaknesses of the method used and raises doubts about the quality of the annotation of the entire dataset and the interpretation that can be made, especially for chemical families and metabolic pathways. As an example, here is a non-exhaustive list of some metabolites for which annotation is questionable:

mustard gas, which is a chemical warfare agent; Phosphoramidate mustard, which is a metabolite of a cytotoxic anticancer drug; 4-Hydroxypropofol, which is a metabolite of an anesthetic; (R)-Amphetamine; Valproic acid glucuronide, a metabolite of an antiepileptic; Valaciclovir, an antiviral; Secobarbital, a sedative; Psilocybin, a hallucinogen; Olamufloxacin, Propofol glucuronide; Dolasetron; Desethylchloroquine; Sparfloxacin; Loperamide; Antibiotic X 14889A; Ketotifen; Tiotropium; Antibiotic X 14889D; Ripisartan; Perindoprilat; Azilsartan medoxomil; Cediranib; S-6-Hydroxywarfarin; Lorcaïnide; Viloxazine; 4'-Azidocytidine; Ibuprofen glucuronide; Mibefradil; Clofibrate; (R)-Ruxolitinib; Terbutaline; Tiagabine; Fluticasone 17 β -Carboxylic Acid; Spironolactone; simvastatin hydroxy acid; Taribavirin; Olodaterol; Protriptyline; Bepridil; Ispinesib; Signazodan; Almorexant; Barbituric acid; Reproterol; cycloguanil; Cetirizine; Flunarizine; Irbesartan; Daprodustat; Imatinib; Lanabecestat; Metoprolol; Chlorambucil; Tafluprost; Ulixertinib; Vardenafil

Thank you for this truly invaluable comment. We appreciate how you have thoroughly engaged with the presented data in detail.

Generally, the concerns raised about the list of incorrectly or questionably annotated metabolites are valid. Nevertheless, these lists rely on the HMDB 2023 entries and undergo logic checks. However, owing to the extensive number of metabolites (HMDB contains > 23k entries) ensuring accuracy for all entries is not always feasible. Considering this weakness, we explicitly mentioned in the manuscript that the annotations are only putative (“*Noteworthy, the identities presented here are putative, but the indicated chemical formulas are highly likely to be accurate.*”). Nonetheless, this method reliably portrays correct chemical formula with a high accuracy, maintaining a mass error of < 1 ppm and an isotopic fine structure of < 300 mSigma.

We have utilized this approach in previous publications, employing level 3 identification which relies on exact mass, fragmentation profiles, and determination of molecular formulas (excluding NMR), resulting in a high level of accuracy. However, assigning metabolite identity solely based on molecular formulas, as in the examples you have presented, is not error-free. Hence, for all the statistically relevant compounds, specifically those namely highlighted in this work (as seen in the results and discussion) we conducted a comprehensive analysis using FT-ICR-MS.

Furthermore, we are fully aware of the existence of a large number of isobaric compounds sharing the same chemical formula. In these cases, MetaboAnalyst selects the first listed metabolite of the annotation list, potentially leading to a questionable assignment of some metabolites. Therefore, we now rectified incorrectly and questionably annotated metabolites by cross-referencing the chemical formulas and exploring plausible alternatives within the annotation list. If feasible alternatives were found in the annotation list, we substituted the incorrectly – or for breath implausible – annotated metabolite.

Some of the **examples** you have listed:

Putative Identity	Chemical Formula	Alternative Metabolite (updated in the raw data)
4-Hydroxypropofol	C12H18O2	(S,Z)-Lyratol acetate
Valproic acid glucuronide	C14H24O8	Octanoylglucuronide
Antibiotic X 14889A	C33H60O8	DG(10:0/PGF1alpha/0:0)
Antibiotic X 14889D	C33H58O7	DG(10:0/0:0/20:3(8Z,11Z,14Z)-2OH(5,6))
Ibuprofen glucuronide	C19H26O8	4,8-Diacetyl-T2-tetrol
simvastatin hydroxy acid	C25H40O6	DG(2:0/20:3(5Z,8Z,11Z)-O(14R,15S)/0:0)
Daprodustat	C19H27N3O6	Glu-Val-Phe

However, when no other metabolite sharing the same chemical formula existed in the annotation list, e.g. for mustard gas or especially for diverse drugs like metoprolol, imitinib, etc., we only listed the chemical formula without assigning a putative identity due to implausible annotation. Some metabolites seem to be questionable, however, those likely are part of the human exposome, which is also included in HMDB. We have thoroughly validated and updated the raw dataset provided in the supplementary data. In order to explain missing metabolites, we have added the following statement in the supplementary data: “*When metabolite annotation seem implausible, only the chemical formula is given without assigning putative identity.”

Moreover, it is important to note that the supplementary dataset exclusively presents the pre-processed raw data. However, the interpretation of this dataset, especially concerning chemical families and metabolic pathways, was carefully conducted using MetaboAnalyst 5.0. As outlined in the manuscript (lines 84-85, 87-88, 261-265), the core metabolome was defined using the HMDB 2023 annotation list, incorporating respective HMDB IDs for the metabolites. However, since not all of those metabolite IDs are deposited in MetaboAnalyst 5.0, only a part of the detected metabolites (>50%) were utilized to determine the core metabolome. Consequently, those annotated metabolites used for the characterization of the breathome were checked for plausibility. Additionally, we specifically included only those classes that accounted for at least 2% of the total hits to enhance the quality of the results based on the dataset used. This ensured that the represented classes had a substantial presence in the dataset, aiming to mitigate the potential impact of incorrectly annotated metabolites that could otherwise skew the representation of chemical families present in the breathome.

Nonetheless, we agree with your comment and therefore address the raised concerns regarding the weaknesses of this method using the HMDB annotation lists in the discussion section (limitations of this study, see 10.).

4. Quantitative aspects: Procedures for data normalization and standardization, especially to enable inter-individual comparisons, are insufficiently detailed. The effects of normalization using the PQN method could be shown in additional data (without vs. with normalization, on the total signal).

Thank you for addressing this concern. We applied the same procedures for data normalization and standardization that were discussed in a previous paper. Brix et al. (2023) successfully demonstrated the impact of PQN, alongside other normalization strategies, on the comparability and quantifiability of the data.

In general, the PQN algorithm typically exerts minimal impact on data exhibiting low biological variation. Similar to the median normalization in MetaboAnalyst, PQN is primarily employed to account for measurement variations over the measurement period, without significantly changing the data structure. The adjustments made by PQN result in only slight changes in intensity. However, neither PQN nor median normalization can effectively compensate for strong batch effects or system drifts. Therefore, prior to processing, the data structure underwent thorough scrutiny to identify such variability, employing methods like PCA analysis to determine the scatter of the quality controls (a pooled sample containing all samples) compared to the samples. A comprehensive comparison of intensities before and after PQN normalization can be found in the cited paper of Brix et al. (2023) (DOI: 10.1021/acs.analchem.3c01380).

5. Signal processing:

(a) How were data from negative controls exploited? Was a filter applied to retain metabolites for volunteers only if their expression was significantly different from that of the blanks? How many metabolites were detected in the blanks?

To validate the results, we measured non-sampled filter material (negative controls) and solvents (blanks). Subsequently, we compared these measurements with the results obtained from samples collected from volunteers participating in the study.

These analyses revealed only minimal traces of metabolites in the filter, which were close to the limit of detection, i.e. 10^6 counts. The same outcome was observed for the measured blanks (containing only the eluent, i.e. 50/50 H₂O/MeOH (v/v)). Overall, the metabolites of our samples showed higher intensities compared to the blanks. All negative controls as well as blanks showed a comparable measurement profile over the entire measurement period, revealing no significant metabolite results.

In total, after applying the same filters as for the samples, we measured 579 and 620 in the blanks (solvent) and negative controls, respectively. However, as we measured these samples to uphold data quality during measurement, we did not discuss the exact values in the manuscript in detail.

Additionally, to uphold the data's quality, we conducted measurements on quality control (QC) samples as an integral part of the data analysis. The QCs are pooled samples encompassing all individual samples, as described by Demetrowitsch et al (2015). Compared to the QCs, no strong metabolite profiles could be measured in the blanks and negative controls. Consequently, we can confidently dismiss the possibility of carryover.

The following sentence was included in the manuscript (see Results):

Additionally, the analysis of negative controls as well as blanks revealed only minimal traces of metabolites in the solvent or on the filter that were close to the detection limit. All negative controls as well as blanks showed a comparable measurement profile over the entire measurement period, revealing no significant metabolites.

(b) What were the results of the analyses of positive controls (spiked samples)?

The positive controls (spiked samples) containing specific selected metabolites exhibited increased intensity upon detection. All targeted metabolites were successfully identified in the data. This analysis validates that, if metabolites are collected on the filter, they can be both extracted and measured using this approach. However, as the FT-ICR-MS is not a quantitative system, the relevance of whether the intensity increased disproportionately or not, is insignificant.

The following sentence was included in the manuscript (see Results):

All positive controls exhibited increased intensities for the spiked metabolites, respectively.

(c) Was a correction applied for different batches?

We used the PQN and median normalization for batch correction, omitting other techniques like the random forest-based approach. This decision was made because the data structure check (before the actual pre-processing) already revealed minimal batch variations. This primarily arises from the relatively short measurement time attributed to the direct injection method and the FT-ICR-MS measurement technique.

The following amendment was made (see Methods, Sample evaluation):

Additionally, different batches were corrected by applying the PQN and median normalization.

(d) How were results from analyses in positive and negative ionization modes managed and concatenated (number of metabolites detected in each mode, degree of overlap, number of metabolites eliminated during the data concatenation step)?

The measured data have been merged during the data pre-processing, which also includes results from ionization in both positive and negative modes.

A heuristic, logical filter was used to merge the data, wherein the count of 'non-detections' of a metabolite is initially compared across all samples and in both ionization modes. The algorithm then selected the values from the ionization mode that had fewer occurrences of zeros ('non-detection'). For instance, based on a sample size of $n=101$, glucose was detected 91 times in positive mode and was not detected 10 times, whereas in negative mode, it was detected 81 times and not detected 20 times. Thus, the algorithm proceeded to retain the values from the positive mode while discarding those from the negative mode. In cases where a metabolite was 'not detected' in an equal number of samples in both modes, the algorithm used the total intensity as a secondary evaluation criteria. The ones with higher intensities were then selected. Consequently, only those results from analyses in positive and negative ionization modes that contained fewer counts for zeros (= non-detections) were used for further data analysis. This merging approach helped us to reduce redundancies in the dataset without losing information.

To explain the merging process, we incorporated the following sentences in the manuscript (methods, sample evaluation):

As part of data merging process, the occurrence of a metabolite was counted in both ionization modes across all samples, respectively. Subsequently, the ionization mode values with the highest number of detections of a specific metabolite across all samples were chosen for further data analysis, while the alternate mode values with lower numbers were disregarded. For instance, if a metabolite was detected 91 and 81 times in the positive and negative mode, respectively, the algorithm proceeded to retain the values from the positive ionization mode. In cases where a metabolite was detected in an equal number of samples in both modes, the total intensity served as a secondary criterion for evaluation.

(e) How were signals corresponding to isotopes treated?

Signals corresponding to isotopes were not thoroughly evaluated. As long as no halogens are involved, isotope signals show significantly lower intensity levels compared to the corresponding main isotope signals. For our non-targeted metabolomics approach, we applied an ultra-high resolution mass spectrometry offering an exceptional mass accuracy (mass error < 1 ppm and isotopic fine structure of < 300 mSigma). Therefore, despite the omission of specific isotope signals in the data evaluation, the precision of high-resolution signals offers precise mass signals specific to the atom composition of the molecular structure, thereby affirming the correctness of the given chemical formula.

Assessing isotopic fine structure, however, appears to be extremely valuable in identifying specific metabolites. Nonetheless, our non-targeted investigation aimed to define the core breathome, serving as a foundation for subsequent analysis in future studies. The additional analysis of isotopic fine structure will be useful for the precise identification of particular breath biomarkers in targeted approaches.

6. Missing values: "To address the missing (zero) values, a fixed value equal to 1/5 of the limit of detection was used." What are the detection limits for each compound, and how were they determined?

Thank you for the comment. The detection limit was not determined for the individual substances, which is according to experiences unusual in metabolomics approach. Instead, it was specified (pre-determined) by the manufacturer of the system, uniformly set at 1 million counts for all metabolites.

We are fully aware that not all metabolites ionize at the same efficiency – some ionizes better than others. Therefore, metabolites are only compared semi-quantitatively between the individual test subjects, rather than comparing metabolites among themselves. This comparison is solely based on sample tests, not on the intensity level of the metabolites.

To make it more clearly for the reader, we have specified the limit of detection in the manuscript:

To address the missing (zero) values, a fixed value equal to 1/5 of the limit of detection (10^6 counts for all metabolites) was used.

7. **Data filter: The complete dataset had 2656 chemical formulas, and a filter was applied to keep only variables present in at least 15% of all samples, resulting in 1139 metabolites. However, in the Excel spreadsheet of results, at least 1897 variables are expressed in more than 15% of volunteers.**

Thank you for your question.

The supplementary dataset presents the pre-processed raw dataset. The difference arises because we uploaded the data in the manuscript's appendix after pre-processing but before cleaning up the missing values. After pre-processing, the data still contains all metabolites that are present in at least 5% of all samples (first filter step).

As part of the ionization merging and the cleaning of the replicates (see methods: sample evaluation), further values are negated, which is why some metabolites are retained that are not in any sample ('NA'). These are then "leftovers" from the data before pre-processing and are eliminated in MetaboAnalyst as a "constant single value". Filtering for the 15% of missing values then takes place in the first step in MetaboAnalyst, significantly reducing the number of metabolites again (second filter step).

Unfortunately, we mistakenly reported 1139 metabolites, but 1138 metabolites were used for further subsequent statistical analyses. We have corrected the number in the manuscript (highlighted in yellow).

To clarify this misinterpretation, we have specified the pre-processed data and filter steps in the following paragraph of the manuscript, with changes highlighted in yellow:

In total, the pre-processed data included 2645 chemical formulas and their putative metabolites that were present in at least 5% of all samples across the aforementioned phenotypic subgroups (see Extended Data). To ensure the identification of the most stable metabolites specific to each subgroup and to minimise the potential interference due to individual variations, further processing focused exclusively on metabolites present in at least 15% of all samples (second filtering step). After applying this filter, 1138 metabolites were selected for the statistical data analysis.

8. **Gender and other covariates' effects: Were the group sizes corresponding to each covariate (tobacco, dietary habits, etc.) evenly distributed between men and women?**

The group sizes corresponding to each covariate were evenly distributed between men and women:

Covariate	Female	Male
Allergies and Intolerances (n=15)	7	8
Smoking (n=10)	3	7
Meat-eater/Flexitarian (n=68)	34	34
Vegetarian (n=28)	13	15
Vegan (n=5)	3	2
Intake of Supplements (n=45)	23	22
Physical Activity Level	45: 23 (regularly) 22 (occasionally)	45: 33 (regularly) 12 (occasionally)
No sports	5	6
Coffee (n=92)	45	47
Alcohol (n=81)	39: 24 (regularly) 15 (occasionally)	42: 29 (regularly) 13 (occasionally)
No alcohol (n=20)	11	9

The distribution of the covariates between the genders can be found in the spreadsheet presenting the detailed sample characteristics (Extended Data). Therefore, the exact number of the group sizes of all covariates are not listed in the manuscript. However, the manuscript now includes the following sentence:

"Noteworthy, other covariates' effects (dietary habits, alcohol consumption, smoking, physical activity, etc.) on the gender-specific metabolic pattern appear to be negligible as the group sizes corresponding to each covariate were evenly distributed between men and women (see Extended Data)."

9. Oral contraceptives' effect: Analyzing based on oral contraceptive use as a binary variable makes little sense without considering the menstrual cycle period, as its effects on the metabolome have been described, and without confirming contraceptive use in the 48 hours preceding the sampling.

Thank you for bringing this matter to our attention. We fully recognize the importance of accounting for the menstrual cycle period when assessing the impact of using oral contraceptives. Disregarding this factor introduces considerable bias into our analysis. Nonetheless, it remains crucial to address the use of hormonal agents in clinical studies to comprehend distinctions in the female population. Despite potential limitations in accuracy, our analysis only indicates that oral contraceptives may indeed have some effects on the metabolome, although these effects are uncertain and require further investigation in future studies. Our primary focus was solely on examining general effects of oral contraceptives. In order to analyse specified effects, a different study approach must consider the exact menstrual cycle period. This would necessitate larger group sizes and a time-dependent analysis, given the considerable variations between menstrual cycles.

In order to consider the mentioned bias, we have also shortened the discussion concerning the use of hormonal agents, as the conclusion drawn may not be definitive. Additionally, we included this major bias as a limitation of the study (see 10.).

10. A "limitations" paragraph should be added to the discussion to address these various points, as well as the likely lack of statistical power to reliably assess the contribution of different covariates.

The discussion has already addressed certain limitations of this study. However, we have included and discussed the following limitations in the discussion:

"A major bias in observed differences between men and women might be attributed to the smaller lung volume in women⁵², potentially resulting in the collection of significantly varying amounts of biological material over 30 exhalations. However, breath sampling was standardized by following the manufacturer's instructions while using the device. The standardized sampling process utilized a plastic bag, which, upon filling, indicates that a sufficient breath volume has passed through the collection filter²⁷. This implies that the potential effect of a smaller lung volume in women may not significantly affect the results."

"However, assessing the effects of oral contraceptives without accounting for the menstrual cycle period introduces a considerable bias into our analysis. We primarily focused on examining general effects of oral contraceptives. In order to analyse conclusive effects, future investigations must consider the exact menstrual cycle period, necessitating larger group sizes and a time-dependent analysis, given the considerable variations between menstrual cycles."

"In general, however, the study might lack statistical power to reliably assess the influence of various covariates, especially due to small group sizes."

"Additionally, it should be noted that the generation of droplets could be highly variable, inter- and intra-individually, between two distinct periods of the day. However, due to the study design, breath samples were collected within a half-hour period. Consequently, the obtained results do not conclusively account droplet variability across different periods. Future studies should aim to compare sampling across multiple periods to effectively assess droplet variability, potentially revealing differing inter- and intra-individual metabolite patterns throughout the day."

"Furthermore, HMDB annotation lists present a substantial weakness of this study. Utilizing these may result in implausible or questionable annotated metabolites (supplementary dataset), likely due to the presence of numerous isobaric compounds that share identical chemical formulas. In these cases, MetaboAnalyst selects the first listed metabolite from the annotation list, potentially leading to a questionable assignment of certain metabolites. Therefore, although the provided chemical formulas are highly accurate, this analytical approach can only suggest putative identities."

Minor Comments

1. The methods indicate that "Participating required a diet protocol." Which protocol?

Thank you for pointing this out. The wording 'diet protocol' might be misleading. The participants were asked to write down the individual intake of any food and beverages over the last 24 hours. Each participant provided a list consisting of meals and drinks they had consumed. The sentence in line 452-453 "Additionally, participating required a diet protocol of the last 24 hours to account for potential influences of the individual nutrition on the metabolomic profile." was changed to:

Additionally, in order to account for potential influences of the individual nutrition on the metabolomic profile, the participants were requested to provide a detailed record of their dietary intake, listing all food and beverages consumed during the 24 hours before taking part in the study.

2. PLS-DA for gender effect: Model metrics (R2Y, Q2Y) should be added to the graph.

We added the model metrics (R2, Q2) to the graph.

3. During what period were the analyses conducted? What were the time intervals between sampling and analysis?

The following sentences were included in the methods (subsection sample preparation) of the manuscript:

The time interval between sampling and analysis was up to six months. After participants were recruited between August 2022 and January 2023, all samples were prepared in batches within four weeks. Subsequently, mass analysis was promptly conducted without any interruptions over a continuous period of two weeks. Prior to analysis, all sample vials were stored at -80 °C."

4. In tables 1 and 2, expression ratios between different groups should be indicated.

The expression ratios (fold change values) between different groups are indicated in the respective tables.

5. The labels of the axes in the graphs are almost illegible.

The labels of the axes in the graphs, especially those of Figure 2, were modified for improved readability.

6. A number of metabolites (DG(11M3/9D3/0:0)DG(9D3/1 ; N-[(4E,8Z)-1,3-dihydroxyoctadeca-4,8-dien-2-yl]hexadecanamide 1-glucoside ; 3-(4-(2-Dimethylamino-1-methylethoxy)phenyl)-1H-pyrazolo(3,4-b)pyridine-1-acetic acid; Austalide F ; Austrobailignan 7; Bakuchiol ; CE(11:1D3) ; Ecadotril ; Glutaminyltyrosine; Pratosartan; Udenafil) are not present in any sample. How can this data be found in the table?

This is because of the filtering and merging process: As part of the ionization merging and the cleaning of the replicates (see methods: sample evaluation), further values are negated, which is why some metabolites are retained in the table that are not detected in any sample ('NA'). These are then "leftovers" from the data before pre-processing and are eliminated in MetaboAnalyst as a constant single value.

These "missing" metabolites were removed from the dataset to avoid misinterpretation. Due to this removal, the number of metabolites was reduced from 2656 to 2645. We corrected the number of the metabolites in the manuscript (highlighted yellow).

7. Consider shortening the discussion section.

Some parts in the discussion were shortened or deleted.

REVIEWERS' COMMENTS:

Reviewer #1 (Remarks to the Author):

The authors have taken care of the individual comments and updated the manuscript. This article will provide useful details on the non-volatile organic compounds in the exhaled breath using DI-FTR-ICR-MS. This resource will be useful to clinical researchers contributing to respiratory medicine, metabolic disorders, and others involved in geriatric research. I am sure it will be very useful for the breath research community.

R2 (withdrawn already before previous round due to unavailability)

Reviewer #3 (Remarks to the Author):

I appreciate the authors for providing thorough responses. All of my comments were successfully addressed and incorporated into the revised version of the manuscript, which, in my opinion, is now suitable for publication.